# Regional $CO_2$ inversions with LUMIA, the Lund University Modular Inversion Algorithm, v1.0

Guillaume Monteil[1] and Marko Scholze[1]

[1]Department of Physical Geography and Ecosystem Science, Lund University, Lund, Sweden

**Correspondence:** Guillaume Monteil (guillaume.monteil@nateko.lu.se)

**Abstract.** Atmospheric inversions are used to derive constraints on the net sources and sinks of $CO_2$ and other stable atmospheric tracers from their observed concentrations. The resolution and accuracy that the fluxes can be estimated with depends, among other factors, on the quality and density of the observational coverage, on the precision and accuracy of the transport model used by the inversion to relate fluxes to observations, and on the adaptation of the statistical approach to the problem studied.

In recent years, there has been an increasing demand from stakeholders for inversions at higher spatial resolution (country scale), in particular in the framework of the Paris agreement. This step up in resolution is in theory enabled by the growing availability of observations from surface in-situ networks (such as ICOS in Europe) and from remote sensing products (OCO-2, GOSAT-2). The increase in the resolution of inversions is also a necessary step to provide efficient feedback to the bottom-up modelling community (vegetation models, fossil fuel emission inventories, etc.). It however calls for new developments in the inverse models: diversification of the inversion approaches, shift from global to regional inversions, improvement in the computational efficiency, etc.

In this context, we developed LUMIA, the Lund University Modular Inversion Algorithm. LUMIA is a python library for inverse modeling, built around the central idea of modularity: it aims at being a platform that enables to construct and to experiment with new inverse modeling setups, while remaining easy to use and maintain. It is in particular designed to be transport model agnostic, which should facilitate isolating the transport model errors from those introduced by the inversion setup itself.

We have constructed a first regional inversion setup using the LUMIA framework to conduct regional $CO_2$ inversions in Europe, using in-situ data from surface and tall tower observation sites. The inversions rely on a new offline coupling between the regional, high resolution FLEXPART Lagrangian particle dispersion model, and the global, coarse resolution TM5 transport model. This test setup is intended both as a demonstration and as a reference for comparison with future LUMIA developments. The aims of this paper are to present the LUMIA framework (motivations for building it, development principles and future prospects) and to describe and test this first implementation of regional $CO_2$ inversions in LUMIA.

# 1 Introduction

The accumulation of greenhouse gases in the atmosphere is the main driver of climate change. The largest contribution of anthropogenic activities to global warming is through the release of fossil carbon (mainly as $CO_2$) to the atmosphere, but other human activities such as land use change (for agriculture, deforestation, etc.) also play a significant role. The climate forcing from this increased greenhouse gases (GHGs) concentration is likely to induce feedbacks through reactions of the terrestrial ecosystems and of the oceans (Stocker et al., 2013). Our capacity to correctly predict climate change, anticipate and mitigate

its effects depends therefore largely on our capacity to model and predict the evolution of carbon exchanges between the atmosphere and other reservoirs.

    One of the main approaches to estimate the land-atmosphere carbon exchanges is through "direct" ecosystem modelling, i.e. using models (numerical or statistical) which simulate, as accurately as possible (given the precision requirements of the simulation) the various carbon exchange processes (respiration, photosynthesis, but also exchanges of carbon between different parts of the plants and the soils, etc.), as a function of environmental parameters (meteorology, soil characteristics, hydrology,

etc.).

    Alternatively, the "inverse" approach infers changes in the $CO_2$ sources and sinks from their observed impact on atmospheric $CO_2$ concentrations. The rationale is that, since the $CO_2$ content in the atmosphere is relatively easy to monitor through direct observations (at very large scales), it is possible to trace back these changes in $CO_2$ concentrations to changes in $CO_2$ sources

and sinks.

    In practice, the direct and inverse approaches are complementary. Ecosystem models can provide detailed estimates of the spatial and temporal variability of the land-atmosphere carbon fluxes, but since they cannot account for the full complexity of the natural processes, they rely on parametrizations, which are not always accurate and can aggregate to large-scale biases. This results in large uncertainties on the total fluxes from ecosystem models Sitch et al. (2015). In contrast, since the total

atmospheric $CO_2$ content is well known, atmospheric inversions can provide robust estimates of the total $CO_2$ fluxes at very large scales (i.e. zonal bands) (Gurney et al., 2002), but they have only poor sensitivity to smaller scales (e.g. Peylin et al. (2013)), the limitation being, ultimately, the density of the observational coverage.

    An atmospheric inverse model typically couples an atmospheric transport model (which computes the relationships between fluxes and concentrations) with an optimization algorithm, whose task is to determine the most likely set of fluxes, within some

prior constraints and given the information from an observation ensemble (in a Bayesian approach). In practice, inversions are complex codes, computationally heavy. The complexity arises in a large part from the necessity to combine large quantities of information from sometimes very heterogeneous datasets (various types of observations, flux estimates, meteorological forcings, etc.). The computational weight depends largely on that of the underlying transport model, which usually needs to be run a large number of times (iteratively or as an ensemble).

In recent years, the availability of observations has grown by orders of magnitude, with the deployment of high-density surface observation networks (such as the Integrated Carbon Observation System, ICOS, in Europe) and the fast developments in satellite retrievals of tropospheric greenhouse gas concentrations (GOSAT, OCO-2, etc.). Meanwhile, the demand for inver-

sions is increasing, in particular from stakeholders such as regional, national or trans-national governments who are interested in country-scale inversions as a means of quantifying their carbon emissions, in connection with emission reduction targets as defined in the Paris agreement (Ciais et al., 2015).

This context puts strain on the existing inverse models. The larger availability of high quality data means that fluxes can be constrained at finer scales, but it also means that models of higher definition and precision must be used. The development of regional inversions (of varying scales) allows in theory an efficient usage of high resolution data while preserving a reasonable computational cost, but comes with specific challenges such as the need for boundary conditions. The demands from various stakeholders (policy makers, bottom-up modellers, medias, etc.) also call for developments in the inversion techniques, with for instance a more pronounced focus on the quantification of anthropogenic sources (Ciais et al., 2015) or the optimization of ecosystem model parameters instead of $CO_2$ fluxes in carbon cycle data assimilation systems (CCDAS) (Kaminski et al., 2013).

To enable such progress in the method and quality of the inversions, it is important to have a robust and flexible tool. The purpose of LUMIA (Lund University Modular Inversion Algorithm) is to be a development platform for top-down experiments. LUMIA was developed from the start as a model-agnostic inversion tool, with a clear isolation of the data stream between the transport model and the optimization algorithm in an interface module. One of the main aims is to eventually allow a better characterization of the uncertainty associated with the transport model. Strong emphasis was put on the usability (low barrier entry code for newcomers, high degree of modularity to allow users to build their experiments in a very flexible way) and sustainability of the code (small, easily replaceable one-tasked modules instead of large multi-option ones).

This paper presents the LUMIA inversion framework, as well as a first application of regional (European) $CO_2$ inversions for Europe. The inversions use in-situ $CO_2$ observations from European tall towers (now part of the ICOS network, see https://www.icos-ri.eu) and rely on a regional transport model based on a new coupling between the FLEXPART Lagrangian particle dispersion model (Seibert and Frank, 2004; Pisso et al., 2019) (regional, high resolution transport) and TM5-4DVAR (Meirink et al., 2008; Basu et al., 2013) (global, coarse resolution). The paper is organized as follows: First, Section 2 presents the LUMIA framework (general principles and architecture). Then Section 3 presents the specific inverse modelling setup used here (including the FLEXPART-TM5 coupling). Sections 4 and 5 present the results from two set of inversions (using synthetic and real observations). Finally, a short discussion summarises the main outcomes of the paper in Section 6.

## 2 The LUMIA framework

### 2.1 Theoretical background

The general principle of an atmospheric inversion is to determine the most likely estimate of a set of variables controlling the atmospheric content and distribution of a tracer (typically sources and sinks, but also initial or boundary conditions), given a set of observations of that tracer's distribution in the atmosphere. The link between the set of parameters to optimize (control vector

x, of dimension $n_\mathbf{x}$) and the observed concentrations (observation vector y, of dimension $n_\mathbf{y}$) is established by a numerical model of the atmospheric transport (and of any other physical process relating the state and observation vectors):

$$\mathbf{y} + \varepsilon_y = H(\mathbf{x} + \varepsilon_x) + \varepsilon_H \tag{1}$$

The observation operator $H$ includes the transport model itself, but also any additional steps needed to express y as a function of x (aggregation/disaggregation of flux components, accounting of boundary conditions and non-optimized fluxes, etc.). The error terms $\varepsilon_y$, $\varepsilon_x$ and $\varepsilon_H$ are respectively the observation error, the control vector error and the model error (see Section 3.4.1). In cases where the observation operator $H$ is linear, $H(\mathbf{x})$ can also be written as $\mathbf{Hx}$, where $\mathbf{H}$ is the Jacobian of $H$. In this case, as an alternative to running a full transport model, it is possible to a pre-compute Jacobian matrix. For simplicity we adopt that syntax throughout the rest of the paper.

In the simplest cases, the system can be solved for x directly, but most often inversions use the Bayesian inference approach: the optimal control vector $\hat{\mathbf{x}}$ is the one that allows the best statistical compromise between fitting the observations and limiting the departure from a prior estimation of the control vector $\mathbf{x_b}$ (accounting for the (prescribed) uncertainties in the observations and the prior). Mathematically, this means finding the vector $\hat{\mathbf{x}}$ that minimizes a cost function $J(\mathbf{x})$ defined (in our case) as

$$J(\mathbf{x}) = \underbrace{\frac{1}{2}\left(\mathbf{x} - \mathbf{x_b}\right)^T \mathbf{B}^{-1}\left(\mathbf{x} - \mathbf{x_b}\right)}_{J_\mathbf{b}} + \underbrace{\frac{1}{2}\left(\mathbf{Hx} - \mathbf{y}\right)^T \mathbf{R}^{-1}\left(\mathbf{Hx} - \mathbf{y}\right)}_{J_{obs}} \tag{2}$$

where the prior ($\mathbf{B}$) and observation ($\mathbf{R}$) error covariance matrices weight the relative contributions to the cost function of each departure from the prior control variables $x_b^i$ and from the observation $y^j$. The optimal control vector $\hat{\mathbf{x}}$ is solved for analytically (for small scale problems) or approximated step-wise (variational and ensemble approaches are most common (Rayner et al., 2018)).

An inversion system is therefore the combination of an observation operator (i.e. transport model, sampling operator, etc.), an inversion technique and a set of assumptions on the prior values of the variables to estimate, their uncertainties and the uncertainties of the observations. Each of these components introduces its own share of uncertainty, which makes the results harder to interpret: which features of the solution are real, and which are introduced by e.g. the transport model, or incorrect assumptions on some uncertainties?

## 2.2 The lumia python package

The LUMIA system is designed with the aim to provide the modularity needed to quantify the impact of the inversion design choices on the inversion results themselves. The strict isolation of the transport model also enables the transport model and the inversion algorithm to evolve independently. Unfortunately, the modularity tends to lead to an increase in the overall complexity of the code (due to the need to develop and maintain generic interfaces), which can end up being counterproductive if it limits

the performances and/or usability of the system. We nonetheless believe that the benefit of a higher modularity outweighs the risks. The potential adverse effects can be mitigated by careful design choices. The code is distributed as a single python package, with the following structure (see also Figure 1):

- The *lumia* folder contains the `lumia` python library, which implements the basic components of the inversion such as data storage (control vector, fluxes, observations, uncertainty matrices) and functions (forward and adjoint transport, conversion functions between fluxes and control vector; cost function evaluation, etc.).

- The *transport* folder contains the `transport` python module, which was used to implement the TM5-FLEXPART transport model coupling, described further in Section 3.3.

- The *src* folder contains the FORTRAN source code for the conjugate gradient minimizer used in the example inversions (see Section 3). Replacing this external code by a native python equivalent is planned.

- The *doc* folder contains documentation, mainly in the form of jupyter-notebooks, and example data and configuration files.

- The *GMDD* folder contains the scripts and configuration files used for producing the results presented further down in this manuscript.

The package can be installed using the standard 'pip' command, which installs `lumia` and `transport` as python modules, which can then be imported from any python script. The `lumia` module itself has a relatively flat hierarchy, which limits the risk that replacing or changing one component prevents the others from working. The implementation of alternative features is preferably carried out via the development of alternative classes, which allows each individual class to remain compact and easy to understand and maintain.

The `lumia` and `transport` modules and their submodules can be used totally independently from the inversion scripts that are provided in the *GMDD* folder. This allows their use in different contexts, such as development, pre/post-processing of the inversion data or during the analysis of the results (and eventually this will help to keep the inversion scripts compact, as they need only to focus on the inversion itself). The scope of the `lumia` library is intentionally vague: it should permit the easy construction of inversion experiments and is primarily designed for it, but our current design choices should not over-constrain the alternative use cases (such as e.g. forward transport model experiments or optimization of land surface model parameters).

## 3   Test inversion setup

Our test inversion setup is designed to optimize the monthly net atmosphere-ecosystem carbon flux (NEE, Net Ecosystem Exchange) over Europe at a target horizontal resolution of $0.5°$, using $CO_2$ observations from the European ICOS network (or similar/precursor sites). Two series of inversions are presented: First, a series of Observing System Synthetic Experiments (OSSEs), using known truth and synthetic observations; then a series of inversions constrained by real observations. All the

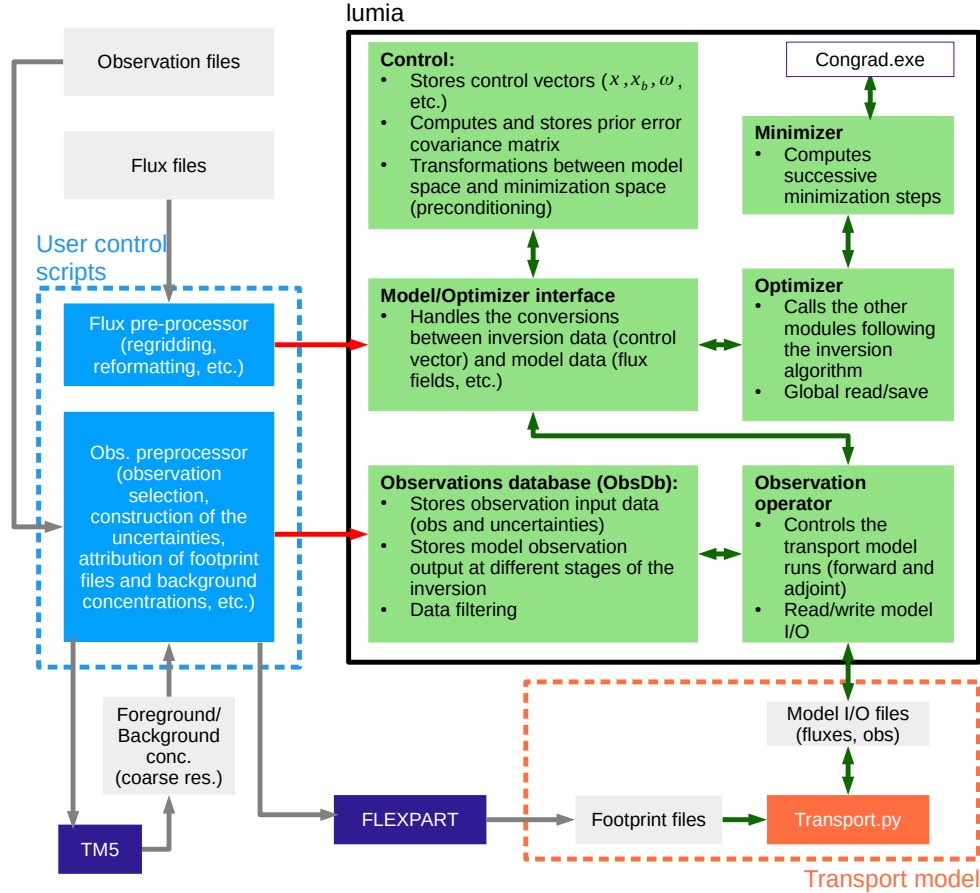

**Figure 1.** Inversion flow diagram. The green boxes represent code that is part of the lumia python module; the orange box shows operations performed by the atmospheric transport model (in our case a simple python script that reads in observations, fluxes and footprints, but a full transport model could be plugged in instead); the blue boxes show code that is typically user- and application-specific (pre-processing of data and main inversion control script). The boxes in grey mark on-disk data, and the boxes in purple show external executables

inversions are performed on a domain ranging from 15°W, 33°N to 35°E, 73°N (illustrated in Figure 2) and cover the year 2011.

The inversions are performed using a variational approach, presented in Section 3.1, with a regional transport model (Section 3.3) that accounts for the impact of regional $CO_2$ fluxes from four process categories (Section 3.5). Sections 3.4 and 3.5 present the fluxes and observations used in the inversions.

### 3.1 Inversion approach

We use a Bayesian variational inversion algorithm, similar to that used in TM5-4DVAR inversions (Basu et al., 2013; Meirink et al., 2008). In a variational inversion, the minimum of the cost function $J(\mathbf{x})$ (Equation 2) is solved for iteratively:

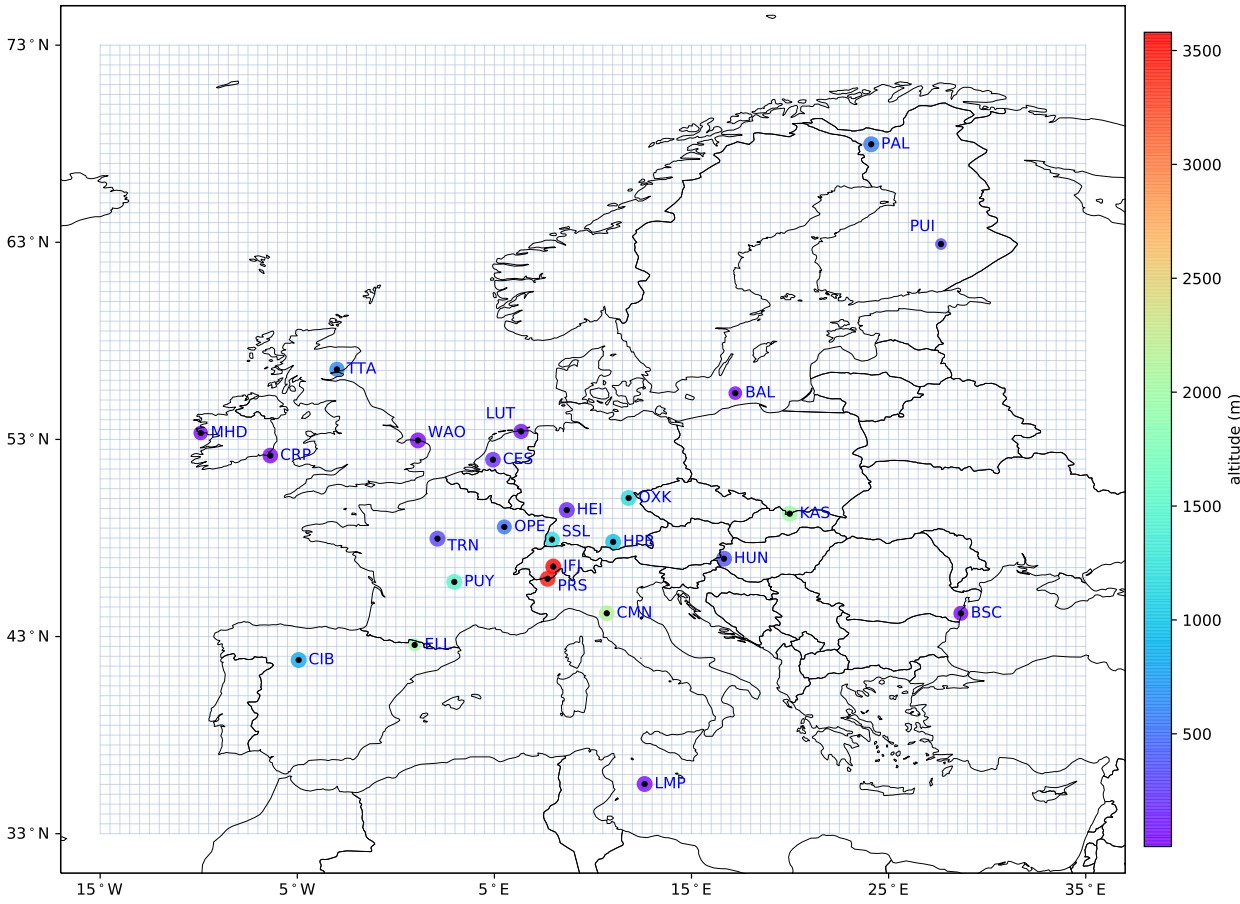

**Figure 2.** Regional inversion domain and location of the observation sites. The area of the dots is proportional to the number of observations available at each site (the actual number of observations is reduced by the filtering described in Section 3.4.4) and their color represents the altitude of the sites.

1. An initial "prior" run is performed to compute the concentrations ($\mathbf{y_m} = \mathbf{Hx_b}$) corresponding to the prior control vector $\mathbf{x_b}$.

2. The local cost function ($J(\mathbf{x} = \mathbf{x_b})$) and cost function gradient ($\nabla_{\mathbf{x}} J(\mathbf{x} = \mathbf{x_b})$) are computed.

3. A control vector increment ($\delta\mathbf{x}$) is deduced from the gradient, and the process is repeated from step 1 (with $\mathbf{x} = \mathbf{x_b} + \delta\mathbf{x}$), until a convergence criterion is reached.

The control vector increments are computed using an external library implementing the Lanczos algorithm (Lanczos, 1950). For efficiency (reduction of the number of iterations) and practicality (reduction of the number of large matrix multiplications)

reasons, the optimization is performed on the preconditioned variable $\omega = \mathbf{B}^{-1/2}(\mathbf{x} - \mathbf{x_b})$ (following Courtier et al. (1994) and similar to the implementation in Basu et al. (2013). Equation 2 then becomes

$$J(\omega) = \frac{1}{2}\omega^T\omega + \frac{1}{2}\left(\mathbf{HB}^{1/2}\omega + \mathbf{d_0}\right)^T \mathbf{R}^{-1}\left(\mathbf{HB}^{1/2}\omega + \mathbf{d_0}\right) \tag{3}$$

with $\mathbf{d_0} = \mathbf{Hx_b} - \mathbf{y}$ the prior model-data mismatches. In this formulation, the cost function gradient is given by

$$\begin{aligned}\nabla_\omega J &= \omega + \mathbf{B^{T/2}H^T R^{-1}}\left(\mathbf{Hx} - \mathbf{y}\right)\\ &= \omega + \mathbf{B^{T/2}}\nabla_\mathbf{x} J_{obs}\end{aligned} \tag{4}$$

The non-preconditioned observational cost function gradient $\nabla_x J_{obs} = \mathbf{H^T R^{-1}}\left(\mathbf{Hx} - \mathbf{y}\right)$ is computed using the adjoint technique (Errico, 1997). The transformation matrix $\mathbf{B}^{1/2}$ is obtained by eigen-value decomposition of $\mathbf{B}$. Note that in this formulation, the inverse of $\mathbf{B}$ (or the square root of its inverse) is actually never needed, making it possible to constrain the

inversion with a non invertible matrix. In practice, the preconditioning adds two extra steps to the algorithm described above: conversion from $\omega$ to $\mathbf{x}$ ($\mathbf{x} = \mathbf{B}^{1/2}\omega + \mathbf{x_b}$) before applying the transport operator (i.e. running the transport model), just before step 1; Conversion from $\nabla_\mathbf{x} J_{obs}$ to $\nabla_\omega J_{obs}$ (just after step 2). The initial preconditioned control vector $\omega$ is filled with zeros and corresponds to $\mathbf{x_0} = \mathbf{x_b}$.

## 3.2 Control vector

The observation operator ($\mathbf{H}$ in Equation 2) groups the ensemble of operations to compute the $CO_2$ concentrations corresponding to a given control vector. It includes the transport model (and its adjoint), but also the steps required to generate transport model parameters based on a given control vector. Indeed, inversions generally don't adjust directly the transport model parameters, but a subset or a construct of them: inversions are generally performed at a lower temporal and/or spatial resolution than the transport, and some of the transport model parameters may be prescribed, such as fluxes from processes that are better

known.

The inversions performed here optimize a monthly offset to the NEE, at the spatial resolution of the regional transport model: the prior control vector slice $\mathbf{x_b}(m)$ containing the gridded offset for the month $m$ is defined as

$$\mathbf{x_b}(m) = \sum_t \mathbf{f}_{0_m}^{nee}(t) \tag{5}$$

with $\mathbf{f}_{0_m}^{nee}$ the prior, 3-hourly NEE for the month $m$.

The reverse operation, i.e. the construction of a 3-hourly NEE field based on a given control vector $\mathbf{x}$, is given by

$$\mathbf{f}_m^{nee} = \frac{\mathbf{x}^m - \mathbf{x}_b^m}{n_t} + \mathbf{f}_{0_{m,nee}}^t \tag{6}$$

with $n_t$ the number of 3-hourly time intervals in that month $m$.

Prescribed $CO_2$ fluxes (anthropogenic, ocean, biomass burning) are stored in memory throughout the inversion and used at each iteration, along with the updated NEE estimate, to generate a new transport model input file.

The adjoint operation corresponding to Equation 6 is simply

$$\mathbf{x}_{adj}(m) = \frac{1}{n_t} \sum_{t=1}^{n_t} \mathbf{f}_{adj_m}^{nee}(t) \tag{7}$$

where $\mathbf{f}_{adj}^{nee}$ is the adjoint NEE flux (computed by the adjoint transport model, described in the next section). Since only the NEE is optimized, the adjoint is zero for the other flux categories.

The adjustment of an offset to the 3-hourly NEE ensures that the amplitude of the daily cycle of NEE remains realistic. This definition of the control vector is by some aspects sub-optimal: in particular, the control vector is un-necessarily large as it contains pixels where NEE is by definition always zero, e.g. in the ocean. However, the aim here was not to obtain the best performing inversion, but a setup easy to develop, test and replicate, that will serve as a base of comparison in future evolutions of the setup. It is however already possible to run LUMIA inversions with more complex configurations, such as variable spatial and temporal resolutions.

## 3.3 Transport model: TM5-FLEXPART coupling

For this first implementation of $CO_2$ inversions with LUMIA, we opted for a regional transport model based on an offline coupling of the TM5 and FLEXPART transport models, following the coupling approach proposed by Rödenbeck et al. (2009). We provide here sufficient information to replicate our setup, but refer to Rödenbeck et al. (2009) for more complete details on the coupling approach.

For each observation $i$, the $CO_2$ concentration $y_m^i$ is defined as the sum of a "foreground" (near-field) and a "background" (far field) contributions:

$$y_m^i = y_{bg}^i + y_{fg}^i \tag{8}$$

– The foreground $y_{fg}^i$ is the contribution of $CO_2$ sources and sinks within a "foreground domain", defined in space as the regional domain shown in Figure 2, and in time as a seven day (in our test setup) time window preceding each observation. This foreground domain is therefore similar in space to all observations, but specific in time to each observation. It accounts for most of the short-term (sub-weekly) variability of the measured $CO_2$ concentrations, but only for a small fraction of the total $CO_2$ (typically less than 10%, see also Figure 3).

– The background $y_{bg}^i$ is the contribution from air entering the regional domain. It groups in fact three contributions: the impact of $CO_2$ present in the atmosphere at the start of the simulation, the impact of $CO_2$ sources and sinks outside the regional domain, and the impact of $CO_2$ sources and sinks within the regional domain but outside the "foreground time window" of the observation. The background is the largest contribution, but it contributes very little to the short-term variability (by the time the air masses from outside the domain reach the observation sites, existing high resolution patterns of $CO_2$ at the regional domain boundaries concentrations would have been dispersed).

We used the global, coarse resolution TM5-4DVAR inversion system to compute the background component of the concentrations. The TM5 setup is described in further details in Section 3.3.2. For the foreground part, we used pre-computed observation footprints computed with the FLEXPART Lagrangian transport model (Section 3.3.1):

$$y_m^i = y_{bg}^i + \sum_i \sum_c \langle \mathbf{K_i}, \mathbf{f_c} \rangle \tag{9}$$

where the footprint $\mathbf{K_i}$ is a vector that stores the sensitivity of the observation $y^i$ to the surface fluxes $\mathbf{f_c}$, and such that $\forall_{\mathbf{f_c}}, \langle \mathbf{K_i}, \mathbf{f} \rangle = K_i(\mathbf{f_c})$, with $K$ the transport model used to compute the footprint $\mathbf{K}$. The footprints and the fluxes are defined on a half-degree, 3-hourly grid, and the fluxes are defined in four categories (NEE, anthropogenic, ocean and biomass burning), described further down in Section 3.5. The fluxes are provided in units of $\mu$mol($CO_2$)/m$^2$/s and the footprints are in s.m$^{-2}$.mol(air)$^{-1}$. The transport model makes no specific distinction between optimized and prescribed fluxes.

The adjoint of the operation represented by Equation 9 is by

$$\mathbf{f}_{adj}^{nee} = \sum_i \mathbf{K}_i^t \delta y^i \tag{10}$$

with $\delta_y^i$ the model-data mismatches weighted by their uncertainties (Section 3.4.1). The adjoint flux field $\mathbf{f}_{adj}$ is calculated only for the flux categories optimized by the inversion (i.e. NEE).

The forward and adjoint transport applications (i.e. Equations 9 and 10) are handled by a separate python executable, called as a subprocess, which acts as a "pseudo transport model" in our inversions. It uses FLEXPART footprints and TM5 background concentrations as input data: neither of the two transport models is directly called during the inversion, and the footprints and backgrounds can in principle be computed using different transport models, or even different methods than those described in Sections 3.3.2 and 3.3.1.

Using pre-computed footprints greatly reduces the computational cost of the inversions, since the forward and adjoint transport model applications simply consist in a series of very simple array operations. This is, however, at the cost of an increase in I/O and storage requirements (one footprint $\mathbf{K}$ must be stored for each observation and is read in at each forward and adjoint iteration). The isolation of the pseudo transport model in a separate executable wasn't necessary from a technical point of view, but it makes its replacement by a full transport model (e.g. a Eulerian model) easier.

### 3.3.1 Observation footprints (FLEXPART)

The footprints ($\mathbf{K}$) were computed using the FLEXPART 10.0 Lagrangian transport model (Seibert and Frank, 2004; Stohl et al., 2010). FLEXPART simulates the dispersion, backwards in time from the observation location, of a large number of virtual air "particles". The footprint $\mathbf{K}_i^{\phi}$ corresponds to the aggregated residence time of the particles released for observation $y^i$, in a given space-time grid box $\phi$ of the regional inversion, and below a threshold altitude layer arbitrarily set to 100m).

The simulations were driven by ECMWF ERA-Interim reanalysis, extracted at a 3-hourly temporal resolution, and on a $0.5° \times 0.5°$ horizontal resolution, on a regional domain ranging from $25°W$, $23°N$ to $45°W$, $83°N$, slightly larger than the inversion grid (in FLEXPART, the output grid, on which the footprints are generated, may be different from the grid of the input meteorological data).

One set of 3-hourly footprints was computed for each observation, up to seven days backward in time (less if all the particles leave the domain sooner). For plain or low altitude sites (see Table 1), the particles were released from the sampling height above ground of the observations. For high altitude sites (around which the orography is unlikely to be correctly accounted for), the particles were released from the altitude above sea level of the observation sites.

The footprints are stored in HDF5 files, following a format described in SI.

### 3.3.2 Background concentrations (TM5)

The background $CO_2$ concentrations ($y^{bg}$ in Equation 9) result from the transport of $CO_2$-loaded air masses from outside the regional inversion domain towards the observation sites. The most straightforward approach to estimate this background influence would be to use a global, coarse-resolution, inversion-optimized $CO_2$ field as boundary condition, and then to transport that boundary condition towards the observation sites using the regional model. However, the risk is that the regional model transports the boundary condition slightly differently than how the global transport model would have done it, which would lead to systematic biases at the observation sites.

An alternative coupling strategy has been proposed by Rödenbeck et al. (2009). Instead of using the regional model to transport the boundary condition, it is the global model from which this boundary condition is extracted that transports it directly to the observation sites. Rödenbeck et al. (2009) used the global transport model TM3 for this, we replicated their approach with the global TM5 transport model Huijnen et al. (2010). The approach consists in the following steps:

1. A global, coarse resolution inversion is performed (with TM5, in our case), constrained by a set of prior $CO_2$ fluxes $f_{apri}^{glo}$ and by a set of surface $CO_2$ observations including both observations outside and inside the regional domain of the LUMIA inversion. The objective of this step is to obtain a set of optimized $CO_2$ fluxes $f_{opt}^{glo}$ that leads to a very realistic atmospheric $CO_2$ distribution in and around the regional domain. The accuracy of the fluxes themselves has less importance.

2. A forward run of TM5 is then used to calculate the $CO_2$ concentrations $y_{TM5}$ corresponding the fluxes $f_{opt}^{glo}$, at all the observation sites within the regional domain.

3. Another forward run of TM5 is used to calculate the foreground concentrations $y_{TM5}^{fg}$, which correspond to the part of $y_{TM5}$ that can results from the portion of the fluxes $f_{opt}^{glo}$ that is within the regional domain. These foreground concentrations are calculated using a modified forward TM5 run, in which 1) the fluxes are same as $f^{glo}$ within the regional domain and are zero outside, and 2) the $CO_2$ concentrations are reset to zero at all time steps everywhere outside the regional domain.

4. The background $CO_2$ concentrations are then given by $y^{bg} = y_{TM5} - y_{TM5}^{fg}$.

Note that here $y_{TM5}^{fg}$ corresponds well to the definition of the foreground concentrations given in Section 3.3: it accounts for the influence of the regional fluxes only as long as the air masses remain within the regional domain. As soon as they leave it, their concentration is reset to zero in the foreground TM5 run. By deduction, this impact of the regional fluxes on the boundary condition is therefore preserved in $y^{bg}$.

The initial global inversion (step 1) was performed using the TM5-4DVAR inversion system (Basu et al., 2013). The TM5 inversions were carried out at a horizontal resolution of $6° \times 4°$ (longitude $\times$ latitude) and 25 hybrid sigma-pressure levels in the vertical, and driven by meteorological fields from the European Centre for Medium-Range Weather Forecasts (ECMWF) ERA-interim reanalysis project Dee et al. (2011). The NEE was optimized in TM5 on a monthly, global $6° \times 4°$ grid, and three additional prescribed $CO_2$ flux categories were transported (fossil fuel, biomass burning and ocean sink). It was constrained by flask observations from the NOAA ESRL Carbon Cycle Cooperative Global Air Sampling Network (Dlugokencky et al., 2019) outside the European domain, and by a subset of the observations used for the regional inversion within the European domain (see Section 3.4 for references, and Table SI1 for a full list of the sites used in that step). For sites with continuous data (i.e. within Europe), the same data filtering as in the LUMIA inversions was used (Section 3.4.4).

The TM5 inversion covers the entire period of the LUMIA inversion, plus six extra months at the beginning and one at the end to limit the influence of the initial condition and to ensure that the background concentrations in the last month of the LUMIA inversion are well constrained by the observations (observations provide important constraints on the fluxes from the preceding month).

Since the focus of the TM5 inversion was only to produce a set of $CO_2$ fluxes ($f_{opt}^{glo}$) that, in step 2, would lead to a realistic global $CO_2$ distribution, the choice of a prior matters a lot less than the selection of observations. For practical reasons, prior fluxes from the CarbonTracker 2016 release were used (Peters et al., 2007): the NEE prior was generated by the CASA model (Potter et al., 1993); fossil fuel emissions spatially distributed according to the EDGAR4.2 inventory (https: //edgar.jrc.ec.europa.eu, last visited: April 2021); biomass burning emissions are based on the GFED4.1s product (Van Der Werf et al., 2017) and the ocean flux is based on the Takahashi et al. (2009) climatology. We refer to the official CarbonTracker 2016 documentation (https://www.esrl.noaa.gov/gmd/ccgg/carbontracker/CT2016, last visited: April 2021) and to references therein for further documentation on these priors.

The total ($y_{TM5}$) and foreground ($y_{TM5}^{fg}$) $CO_2$ concentrations were saved continuously (every 30 minutes) at the coordinates of each of the observation sites used in the LUMIA inversion. The concentrations were sampled at the altitude (above sea level)

of the observation sites, but also as vertical profiles between the surface (as defined in the TM5 orography) and 5000 m.a.s.l (with a vertical resolution of 250 m), which were used to construct a part of the observation uncertainties.

## 3.4 Observations and observational uncertainties

Observations from the GLOBALVIEWplus 4.2 obspack product were used in the inversions (Cooperative Global Atmospheric Data Integration Project, 2017). For the year 2011, the product includes observations from 26 sites within our regional domain (in addition to observations from mobile platforms, which were not used). Continuous observations are available at 18 of these 26 sites and nine sites are high altitude. Most of these observation sites are now part of the European ICOS network. A list of the sites (coordinates, observation frequency, sampling height and data provider) is provided in Table 1, and the location of the 315 observations is also reported in Figure 2.

### 3.4.1 Observation uncertainties

The observation uncertainty matrix ($\mathbf{R}$) accounts for both the measurement uncertainties ($\varepsilon_{obs}$) and the model representation uncertainty ($\varepsilon_H$, i.e. the incapacity of the model to represent perfectly well the observations, even given perfect fluxes). In theory, the diagonal of the matrix stores the absolute total uncertainty associated to each observation while the off-diagonals 320 should store the observation error correlations. In practice, these correlations are difficult to quantify, and the size of the matrix would anyway make it impractical to invert. The off-diagonals are therefore ignored in our system (as in most similar inversion setups) and the observation uncertainty is stored in a simpler observation error vector, $\varepsilon_y$.

Our inversion system uses an observation operator that decomposes the background and foreground components of the $CO_2$ mixing ratio, therefore the model uncertainty can itself be decomposed in foreground and background uncertainties:

$$325 \quad \varepsilon_y = \sqrt{\max\{\varepsilon_{obs}, \varepsilon_{obs}^{min}\}^2 + \varepsilon_{bg}^2 + \varepsilon_{fg}^2} \qquad (11)$$

The instrumental error ($\varepsilon_y$) is provided by the data providers for most of the observations, and typically ranges between 0.1-0.7 ppm (see Figure 3). We enforced a minimum instrumental error ($\varepsilon_{obs}^{min}$) of 0.3 ppm for all the observations.

The model representation error cannot be formally quantified, as this would require knowing precisely the $CO_2$ fluxes that the inversion is attempting to estimate. One can, however, assign representation error estimates (foreground and background) 330 based, in particular, on assumptions of situations that would normally lead to a degradation of the model performances (for instance late-night/early-morning observations, with a development of the boundary layer that may not be well captured by the model, or observations in regions with a complex orography). Transport model comparisons can also provide representation error estimates based on the difference in their results.

### 3.4.2 Foreground model uncertainties

As described in Section 3.3.2, the TM5 simulation used for computing background $CO_2$ concentrations, also computes the foreground concentrations at each observation site. We performed a forward transport simulation with the regional transport model in LUMIA, using both the background concentrations and the foreground fluxes from that TM5 simulation, so that the two simulations differ only by their regional transport model. A comparison between the concentrations computed by the two models is shown in Figure 4. The bias between the two models is very small during the summer months (it is below 0.2 ppm from April to September, and goes as low as 0.01 ppm in July), but rises during the winter months (up to 1.45 ppm in November). The mean average difference between the two simulations is also much larger in winter: it ranges from 0.82 ppm in September to 4.3 ppm in November, with a yearly average of 3.3 ppm.

This comparison is not a formal performance assessment of either TM5 or of the FLEXPART-based transport used in LUMIA, and in particular the bias should be interpreted with care as the sign of the total net foreground flux changes during the year (which mechanically leads to a change of the sign of the bias). Nonetheless, it provides an indication on the order of magnitude of the foreground model transport errors. We use the absolute differences between the two models as a proxy for $\varepsilon_{fg}$.

### 3.4.3 Background model uncertainties

Background concentrations are expected to be accurately estimated by the global TM5 inversion when the dominant winds are from the West and that any signal from a strong point $CO_2$ source or sink has had time to dissipate along the air mass trajectory over the Atlantic Ocean. In less favourable conditions, there can be entries of less well-mixed air inside the domain, in particular in case of Easterly winds or in events of re-entry of continental air that would have previously left the domain. These events are less likely to be well captured by the TM5 inversion and should be attributed a higher uncertainty.

There is no perfect and easy way to detect these events, but one of their consequences would be a less homogeneous background $CO_2$ distribution around the observation sites when they occur. As part of the TM5 simulation, vertical profiles of background concentrations were stored for each observation (from the surface to 5000 m.a.s.l, at a 250m vertical resolution). We set the background uncertainty of each observation ($\varepsilon_{bg}$) to the standard deviation of its corresponding background $CO_2$ vertical profile. $\varepsilon_{bg}$ is on average 0.36 ppm, one order of magnitude lower than $\varepsilon_{fg}$, and it is also more constant (it ranges between 0.01 and 3.6 ppm). Note that these statistics are computed before the observation selection procedure, described in the following section. The different components of the observation uncertainty are compared in Figure 3 for two representative sites.

### 3.4.4 Observation selection

The inversions are performed on a subset of the observations included in the obspack product. Only observations for which the transport model simulation is expected to result in accurate concentrations are kept. In practice, one of the main difficulties of transport modelling is to correctly compute the mixing of air in the lower troposphere below the boundary layer. The lowest

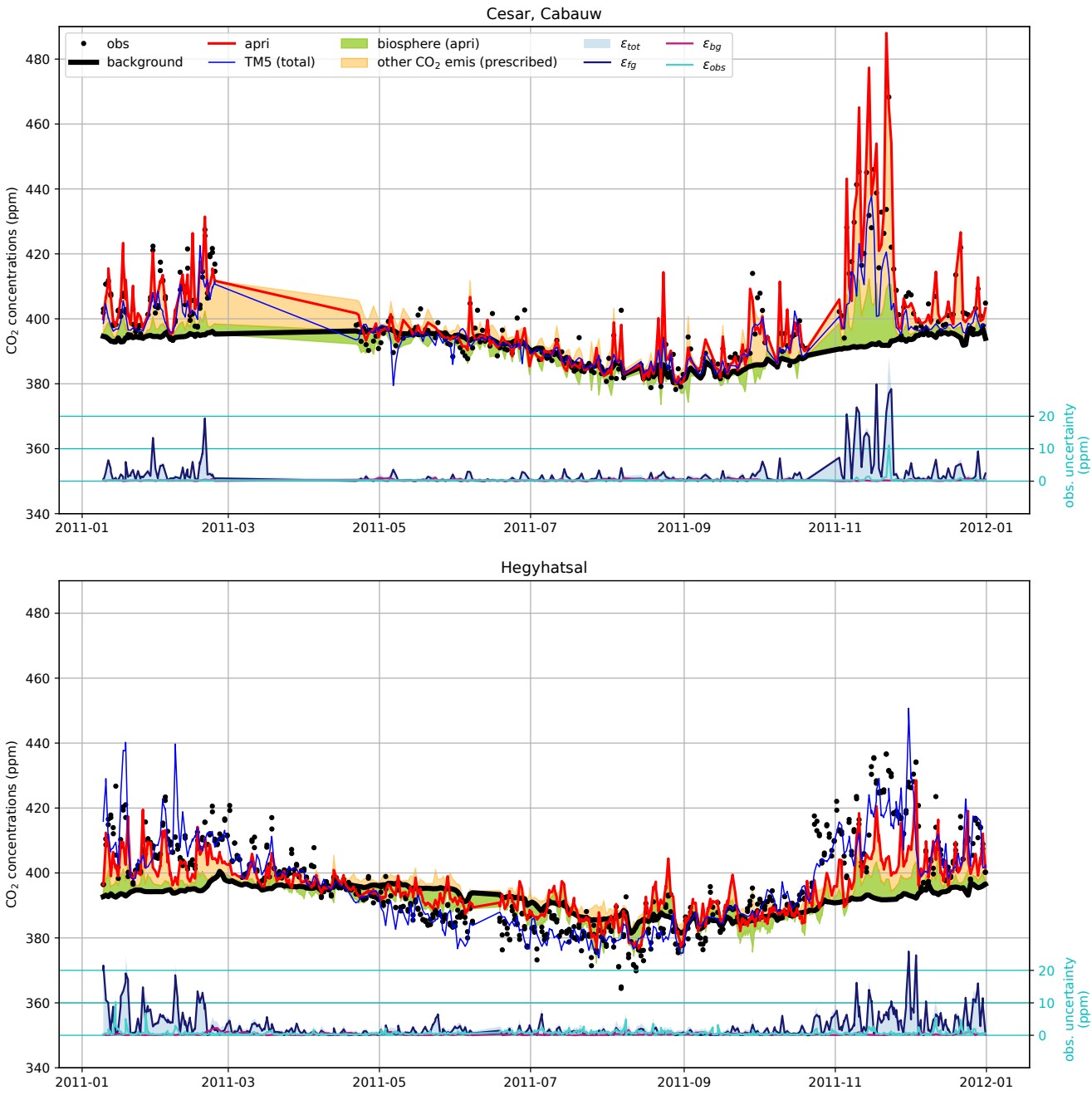

**Figure 3.** Decomposition of the modelled mixing ratio and of the observation uncertainties at two sites (Cabauw, The Netherlands and Hegyhatsal, Hungary). The "TM5 total" line is the concentration computed in the coarse resolution TM5 inversion from which the background (thick black line) is extracted. The LUMIA prior concentration is shown in red and the green and orange shaded areas show respectively the contribution of the prior biosphere flux and of the other $CO_2$ fluxes to the difference between that prior and the background. The lower series of lines in each plot (with y-axis on the right) shows the total observation uncertainty (blue shaded area), and the contributions of the foreground, background and observational uncertainties.

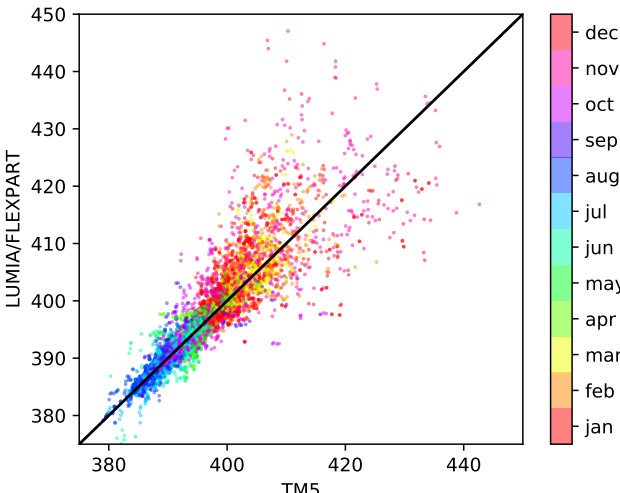

**Figure 4.** LUMIA ($CO_2$ concentrations obtained with TM5-FLEXPART vs. with TM5, using the $CO_2$ fluxes used as prior of the TM5 inversion (background). The color of the dots show the observation month.

model representation error is expected for observations that are either within the boundary layer when it is most developed (in the afternoon), or well above the boundary layer for high-altitude sites (during the night). For each site with continuous observations, we selected only observations sampled during the time range for which the model is expected to perform the best. The time ranges are based on the "`dataset_time_window_utc`" flag in the metadata of the observation files from the obspack. For sites with discrete sampling, all observations were used.

### 3.5 Prior and prescribed fluxes

In addition to the Net Ecosystem Exchange (NEE, net atmosphere-land $CO_2$ flux) that is optimized in the inversions, the transport model also account for anthropogenic $CO_2$ emissions (combustion of fossil fuels, bio fuels and cement production), for biomass burning emissions (large scale forest fires) and for the ocean-atmosphere $CO_2$ exchanges.

The NEE prior is taken from simulations of the LPJ-GUESS and ORCHIDEE vegetation models: in the OSSEs (Section 4) ORCHIDEE fluxes are used as prior and LPJ-GUESS fluxes are used as truth, while in inversions against real data LPJ-GUESS fluxes are used as prior. Both vegetation models provide 3-hourly fluxes, on a horizontal $0.5° \times 0.5°$ grid.

LPJ-GUESS (Smith et al., 2014) is a dynamic global vegetation model (DGVM), which combines process-based descriptions of terrestrial ecosystem structure (vegetation composition, biomass and height) and function (energy absorption, carbon and nitrogen cycling). The vegetation is simulated as a series of replicate patches, in which individuals of each simulated plant functional type (or species) compete for the available resources of light and water, as prescribed by the climate data. The model is forced using the WFDEI meteorological data set (Weedon et al., 2014) and produces 3-hourly output of gross and net carbon fluxes.

| Code | Name | Lat (°E) | Lon (°N) | Alt (m.a.s.l) | Intake Height (m.a.g.l) | Nobs | Time range (h) | sets | Data provider |
|---|---|---|---|---|---|---|---|---|---|
| BAL | Baltic Sea | 55.35 | 17.22 | 3 | 25 | 53 | all | P | 1 |
| BSC | Black Sea, Constanta | 44.18 | 28.66 | 0 | 5 | 17 | all | P | 1 |
| CES200 | Cesar, Cabauw | 51.97 | 4.93 | -1 | 200 | 306 | 11-15 | P | 2 |
| CIB005 | Centro de Investigacion de la Baja Atmosfera (CIBA) | 41.81 | -4.93 | 845 | 5 | 49 | * | A | 1 |
| CMN | Mt. Cimone Station | 44.18 | 10.70 | 2165 | 12 | 549 | 23-3 | A | 3 |
| CRP | Carnsore Point | 52.18 | -6.37 | 9 | 14 | 589 | 12-16 | P | 4 |
| ELL | Estany Llong | 42.57 | 0.95 | 2002 | 3 | 8 | 11-15 | A | 5 |
| HEI | Heidelberg | 49.42 | 8.67 | 116 | 30 | 632 | 11-15 | P | 6 |
| HPB054 | Hohenpeissenberg | 47.80 | 11.02 | 936 | 54 | 47 | all | A | 1 |
| HUN115 | Hegyhatsal | 46.95 | 16.65 | 248 | 115 | 685 | 11-15 | P | 7 |
| JFJ | Jungfraujoch | 46.55 | 7.99 | 3570 | 10 | 461 | 23-3 | A | 8 |
| KAS | Kasprowy Wierch | 49.23 | 19.98 | 1989 | 5 | 481 | 23-3 | A | 9 |
| LMP005 | Lampedusa | 35.52 | 12.62 | 45 | 5 | 35 | all | P | 1 |
| LMP008 | Lampedusa | 35.52 | 12.62 | 45 | 8 | 418 | 10-14 | PA | 10 |
| LUT | Lutjewad | 53.40 | 6.35 | 1 | 60 | 289 | 11-15 | P | 11 |
| MHD024 | Mace Head | 53.33 | -9.90 | 5 | 24 | 352 | 12-16 | P | 12 |
| OPE120 | Observatoire Perenne de l'Environnement | 48.56 | 5.50 | 390 | 120 | 405 | 11-15 | P | 12 |
| OXK163 | Ochsenkopf | 50.03 | 11.81 | 1022 | 163 | 48 | all | A | 1 |
| PAL | Pallas-Sammaltunturi | 67.97 | 24.12 | 565 | 5 | 654 | 22-2 | PA | 13 |
| PRS | Plateau Rosa Station | 45.93 | 7.70 | 3480 | 10 | 445 | 23-3 | A | 14 |
| PUI | Puijo | 62.91 | 27.65 | 232 | 84 | 170 | 11-15 | P | 13 |
| PUY010 | Puy de Dome | 45.77 | 2.97 | 1465 | 10 | 409 | 23-3 | A | 12 |
| PUY015 | Puy de Dome | 45.77 | 2.97 | 1465 | 15 | 141 | 23-3 | A | 12 |
| SSL | Schauinsland | 47.92 | 7.92 | 1205 | 12 | 625 | 23-3 | A | 15 |
| TRN180 | Trainou | 47.96 | 2.11 | 131 | 180 | 539 | 11-15 | P | 12 |
| TTA | Tall Tower Angus | 56.56 | -2.99 | 400 | 222 | 435 | 12-16 | PA | 16 |
| WAO | Weybourne, Norfolk | 52.95 | 1.12 | 20 | 10 | 1078 | 12-16 | P | 17 |
| WES | WES | 54.93 | 8.32 | 12 | 0 | 1377 | 11-15 | P | 18 |

**Table 1.** Observation sites used in the inversions. The "sets" column refers to the site selection applied in inversions SA/SP and RA/RP, where set "P" includes only low-altitude sites and set "A" includes mostly high-altitude sites. Data providers: 1:Cooperative Global Atmospheric Data Integration Project (2017); 2:Vermeulen et al. (2011); 3:Ciattaglia et al. (1987); 4:D. Dodd (EPA Ireland); 5:J.A. Morgui and R. Curcoll (ICTA-UAB, Spain); 6:Hammer et al. (2008); 7:Haszpra et al. (2001); 8:Uglietti et al. (2011); 9:Rozanski et al. (2014); 10:A. G. di Sarra (ENEA, Italy); 11:van der Laan et al. (2009); 12:Yver et al. (2011); 13:Hatakka et al. (2003); 14:F. Apadula (RSE, Italy); 15:Schmidt (2003); 16:Ganesan et al. (2015); 17:Wilson (2012); 18:K. Uhse (UBA, Germany)

ORCHIDEE is a global process-based terrestrial biosphere model (initially described in Krinner et al. (2005)) that computes

carbon, water and energy fluxes between the land surface and the atmosphere and within the soil-plant continuum. The model computes the Gross Primary Productivity with the assimilation of carbon based on the Farquhar et al. (1980) model for C3 plants and thus accounts for the response of vegetation growth to increasing atmospheric $CO_2$ levels and to climate variability. The land cover change (including deforestation, regrowth and cropland dynamic) were prescribed using annual land cover

| Category | Product | Original resolution | Data provider | Total (min/max) flux (PgC/year) |
|----------|---------|---------------------|---------------|----------------------------------|
| Biosphere | LPJ-GUESS | $0.5° \times 0.5°$; 3-hourly | Lund University | -0.33 (-2.65 / 1.85) |
| Biosphere | ORCHIDEE | $0.5° \times 0.5°$; 3-hourly | LSCE (P. Peylin, pers. comm) | -0.28 (-3.73 / 2.14) |
| Fossil | EDGARv4.3 | $0.1° \times 0.1°$; hourly | ICOS-CP + JRC | 1.53 |
| Ocean | CarboScopev (oc_v1.7) | $5° \times 3.83°$; daily | Rödenbeck et al. (2013) | -0.11 (-0.05 / 0.01) |
| Fires | GFEDv4 | $0.5° \times 0.5°$; 3-hourly | Van Der Werf et al. (2017) | 0.01 |

**Table 2.** Prior and prescribed $CO_2$ fluxes. Min/Max values are provided for the fluxes that have both positive and negative components, and correspond to the minimum and maximum values of the 3-hourly flux aggregated over the entire domain, in PgC/year.

maps derived from the Harmonized land use data set (Hurtt et al., 2011) combined with the the ESA-CCI land cover products.
The net and gross $CO_2$ fluxes used for this project correspond to the one provided for Global Carbon Project inter-comparison
(Le Quéré et al., 2018) with a model version that was updated recently (Peylin et al., in preparation).

Fossil fuel emissions are based on a pre-release of the EDGARv4.3 inventory for the base year 2010 (Janssens-Maenhout
et al., 2019). This specific dataset includes additional information on the fuel mix per emission sector and thus allows for
a temporal scaling of the gridded annual emissions for the inversion year (2011) according to year-to-year changes of fuel
consumption data at national level (bp2, 2016), following the approach of Steinbach et al. (2011). A further temporal disaggregation into hourly emissions is based on specific temporal factors (seasonal, weekly, and daily cycles) for different emission
sectors (Denier van der Gon et al., 2011).

The ocean-atmosphere flux is taken from the Jena CarboScope v1.5 product, which provides temporally and spatially resolved estimates of the global sea-air $CO_2$ flux, estimated by fitting a simple data-driven diagnostic model of ocean mixed-layer
biogeochemistry to surface-ocean $CO_2$ partial pressure data from the SOCAT v1.5 database (Rödenbeck et al., 2013).

A biomass burning flux category was also included in the inversion, based on fluxes from the Global Fire Emission Database
v4 (Giglio et al., 2013). In our European domain biomass burning emissions are negligible compared to the other $CO_2$ emission
sources, however, we include it for completeness.

All fluxes are regridded on the same $0.5° \times 0.5°$, 3-hourly resolution (by simple aggregation or re-binning). A summary of
the prior fluxes sources, original resolution and yearly totals is provided in Table 2.

### 3.5.1 Prior uncertainties

The background error covariance matrix ($\mathbf{B}$ in Equation 2) is constructed following the "correlation length" approach used
in many other inversion systems (e.g. Houweling et al. (2014); Thompson et al. (2015); Chevallier et al. (2005)): The error
covariance between control vector elements $\mathbf{x_1}$ and $\mathbf{x_2}$ at grid cells with coordinates $p_1 = (i1, j1, t1)$ and $p_2 = (i2, j2, t2)$ is
defined as:

$$cov(\mathbf{x_1}, \mathbf{x_2}) = \sigma_{\mathbf{x_1}}^2 . \sigma_{\mathbf{x_2}}^2 e^{-(d(p_1, p_2)/L_h)^2} e^{-|t_1 - t_2|/L_t} \tag{12}$$

where $\sigma_{\mathbf{x_1}}^2$ and $\sigma_{\mathbf{x_2}}^2$ are the variances assigned to the prior monthly NEE at coordinates $p_1$ and $p_2$, and $L_h$ and $L_t$ are correlation lengths, which define how rapidly the correlation between two components drops as a function of their distance in time and space.

The true uncertainty of the prior fluxes ($\sigma_{\mathbf{x}}^2$) is difficult to evaluate and is therefore constructed on reasonable but arbitrary assumptions. We tested several approaches, further discussed in Section 4: 1) Scaling the uncertainties linearly to the absolute monthly flux; 2) Scaling the uncertainties to the absolute net 3-hourly flux (and then cumulating these 3-hourly uncertainties to the monthly scale); 3) Enforcing constant monthly uncertainties throughout the year, at the domain-scale.

## 3.6 Inversions performed

We performed two set of inversions, listed in Table 3. The first set consists of Observing System Simulation Experiments (OSSEs) and is design to assess the theoretical performance of the inversions, in the absence of transport model errors. The second test consists of inversions using real observations/

In the OSSEs, the LPJ-GUESS NEE dataset was taken as an arbitrary truth, and a dataset of synthetic pseudo-observations was generated at the times and locations of the actual observations listed in Table 1, by forward propagation of the "true" NEE flux with the transport model (including also the contributions of non-optimized fluxes listed in Section 3.5). Random perturbations were then added, to mimic the measurement error ($y = y^{truth} + \mathcal{N}(0, \sigma_y^2)$, with $\sigma_y^2$ the uncertainty of each observation as defined in the matrix $\mathbf{R}$).

The OSSEs use this set of pseudo-observations as observational constraints and the ORCHIDEE NEE dataset as prior. The reference OSSE, SRef, uses a prior error covariance matrix ($\mathbf{B}$) constructed with prior uncertainties set to 25% of the absolute prior value ($\sigma_{\mathbf{x_b}}^2 = 0.25|\mathbf{x_b}|$) and with covariances constructed from a horizontal correlation length ($L_H$) of 200 km and a temporal correlation length ($L_t$) of 30 days. In the sensitivity tests (listed also in Table 3) we vary the correlation lengths (SC.100 and SC.500), the prescribed prior uncertainties (SE.3H, SE.3Hcst, SE.x2) and the extent of the observation network (SO.A, SO.P).

The second set is essentially identical to the OSSEs, except that it is using real observations, and the LPJ-GUESS flux dataset as a prior.

## 4 OSSEs

We first analyze the capacity of the reference OSSE SRef to reconstruct various characteristics of the "true" LPJ-GUESS NEE fluxes (monthly and annual NEE budget, aggregated at spatial scales ranging from the entire domain down to single pixels). Then we analyze the results of the other OSSEs to test how sensitive the results are to a range of reasonable assumptions in the inversion settings.

| Simulation | Prior | Observations | $\sigma_{\mathbf{x}}^2$ | $L_h$ | $L_t$ |
|---|---|---|---|---|---|
| SRef | ORCHIDEE | pseudo | 25% of monthly prior | 200 km | 1 month |
| SC.500 | - | - | - | 500 km (exp) | - |
| SC.100 | - | - | - | 100 km (exp) | - |
| SE.x2 | - | - | 50% of monthly prior | 200 km | - |
| SE.3H | - | - | 12.5% of 3-hourly prior | 200 km | - |
| SE.3Hcst | - | - | 12.5% of 3-hourly prior, scaled to the same total value every month | - | - |
| SO.P | - | pseudo (set P) | - | 200 km | - |
| SO.A | - | pseudo (set A) | - | 200 km | - |
| RRef | LPJ-GUESS | real | 25% of monthly prior | 200 km | 1 month |
| RC.500 | - | - | - | 500 km (exp) | - |
| RC.100 | - | - | - | 100 km (exp) | - |
| RE.x2 | - | - | 50% of monthly prior | 200 km | - |
| RE.3H | - | - | 12.5% of 3-hourly prior | 200 km | - |
| RE.3Hcst | - | - | 12.5% of 3-hourly prior, scaled to the same total value every month | - | - |
| RO.P | - | real (set P) | - | 200 km | - |
| RO.A | - | real (set A) | - | 200 km | - |

**Table 3.** List of inversion experiments performed. The restricted observation sets A and P are reported in Table 1.

## 4.1 Reference inversion (SRef)

Figure 5 shows monthly and annual time series of NEE and NEE error (with respect to the prescribed truth) aggregated over the entire domain.

At the domain-scale, the prior estimate for the annual NEE is very close to the "truth" (respectively -0.28 and -0.34
PgC/year), but the amplitude of the seasonal cycle in the prior is more than double that of the truth, with monthly NEE ranging from +0.26 PgC/month in October to -0.66 PgC/month in June in the prior, compared to +0.08 PgC/month (in January) to -0.29 PgC/month (in May) in the truth. In total, the absolute prior error slighly exceeds 3 PgC/year and peaks in June and July and is the lowest in December-February.

The inversion improves the estimation of the seasonal cycle at the domain scale, with a seasonal cycle amplitude reduced to
450 a range of -0.36 PgC/year (May) to +0.16 PgC/year (December), much closer to the truth, and the absolute error is reduced by nearly 40% to 1.87 PgC/year. However since the positive flux corrections in the summer months largely exceed the negative corrections from September to April, this results in a strong degradation of the annual European NEE estimate, with a near-balanced posterior flux of -0.05 PgC/year. This is largely because the prior was already a very good estimate of the annual NEE, but is somewhat counter-intuitive to the assumption that a reduction of the errors would necessarily mean an improvement of
455 the flux estimates at large spatial and temporal scales.

Figure 6 illustrates the spatial distribution of the error reduction. While the largest prior errors are found north of the Black Sea and in North-Africa, the error reduction is rather homogeneous, except for North-Africa and Turkey (which are not really constrained by the observation network), and some patches in Western Europe (mainly in the UK, but also in Ireland, France

and the Benelux) where the error actually increases. In total, these localized error enhancements amount to 0.16 PgC/year
(lower panel of Figure 5). These isolated occurrences of error enhancements are not a sign of malfunction of the inversion
system, but they highlight its limitations: they result from attributions of flux corrections to the incorrect grid cells, which can
happen if the resolution of the inversion is not adapted to the constraints provided by the observation network (i.e. smoothing
and aggregation errors, as defined in Turner and Jacob (2015)).

Although our control vector contains the flux estimates at the native spatial resolution of the transport model, the effective
resolution of the inversion is further constrained by the covariances contained in the prior error-covariance matrix $\mathbf{B}$. Further-
more, the fluxes are only optimized monthly, while the actual prior error varies at a 3-hourly resolution. Since the network
is not homogeneous, the choice of the inversion is necessarily a compromise, we test its impact in the next Section. The low
temporal resolution was mostly chosen for practical reasons, as it made development and testing easier, but it corresponds to
a standard practice in regional inversions, which tend to either optimize the fluxes at a low temporal resolution (Thompson
and Stohl, 2014; van der Laan-Luijkx et al., 2017), or to impose temporal covariance length on the order of one month (e.g.
Broquet et al. (2011); Kountouris et al. (2018)).

## 4.2 Sensitivity tests

The total NEE flux, absolute error and error increase are shown in Figure 5, for the individual sensitivity experiments at the
annual scale and as an ensemble shape for the monthly scale (the monthly-scale results of the individual simulations can be
found in Figure SI2).

### 4.2.1 Sensitivity to the error distribution

Inversions SE.3H, SE.3Hcst and SE.x2 were designed to test the impact of the prescribed prior uncertainty vector (e.g. diagonal
of $\mathbf{B}$) on the inversion:

- In SE.3H, the prior uncertainty is set proportional to the sum of the uncertainties on the 3-hourly fluxes: $\sigma_{\mathbf{x_b}} = \frac{0.13}{T} \sum_t^T |f_t|$.
  This avoids the situation where GPP and respiration are significant but compensate each other, leading to a near zero
  NEE as well as a near zero prior uncertainty, which can happen when the prior uncertainty is calculated following the
  approach used in SRef. The factor 0.13 was chosen to lead to a total annual uncertainty comparable to that of SRef. This
  leads to an overall redistribution of the uncertainties from the winter to the summer period, which is closer to the actual
  distribution of differences between the prior and truth fluxes (see Figure SI1).

- In SE.3Hcst, the prior uncertainty is computed as in SE.3H, but it is then scaled monthly, so as to lead to a flat distribution
  of the uncertainties across the year.

- In SE.x2, the prior uncertainty is simply doubled compared to SRef.

SE.3Hcst leads to an improved value of the annual budget of NEE at the domain scale, but this is due to a poorer estimation
of the summer fluxes (since the uncertainty is lower in summer, the inversion sticks more to its prior). On the contrary, SE.3H

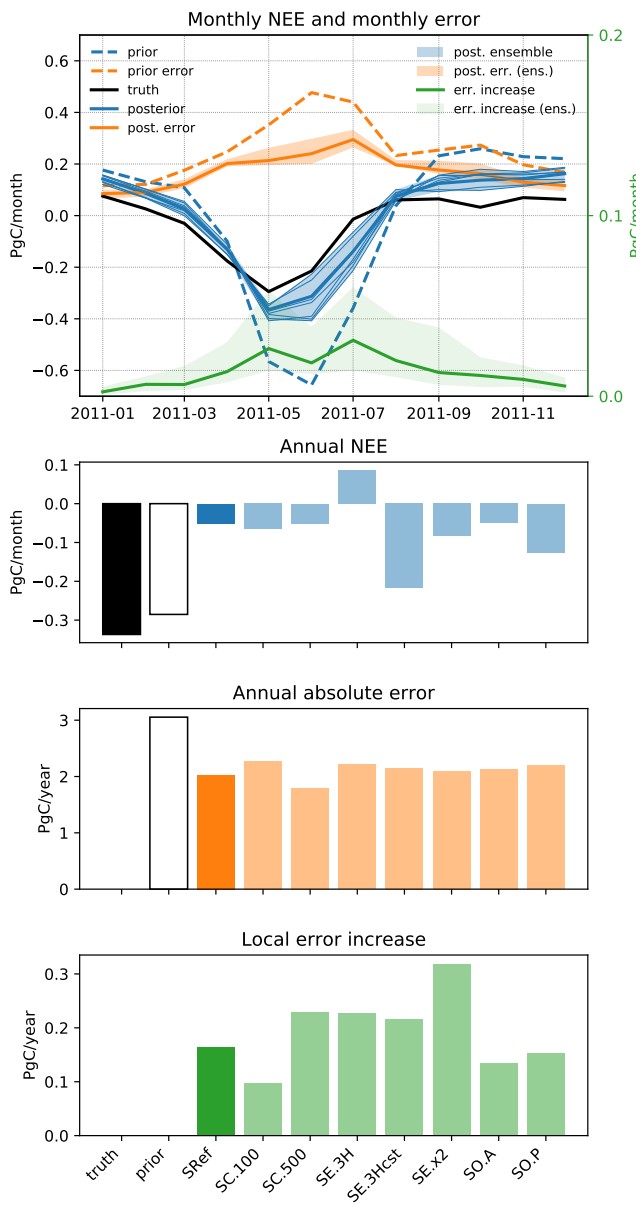

**Figure 5.** Upper row, left axis: Monthly prior NEE (dashed blue line), true NEE (solid black line), posterior NEE (blue), absolute prior error (dashed orange line) and posterior error (orange) in the OSSEs; Upper row, right axis: Total error increase (i.e. positive component of the error reduction, green). The SRef inversion is shown as solid lines, the set of sensitivity tests is shown as a shaded area (prior/post. ensemble). Second, third and fourth rows: same variables, but aggregated annually.

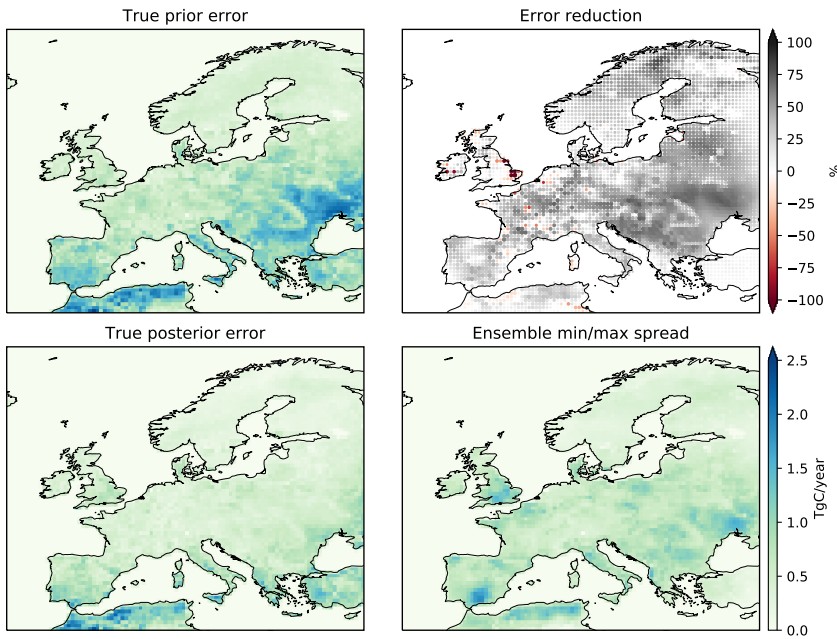

**Figure 6.** Left column: prior (top) and posterior (bottom) total error (with respect to the prescribed truth); Upper-right: percentage error reduction. The size of the dots is proportional to the size of the flux correction; Bottom-right: spread of the posteriors in throughout the ensemble of sensitivity tests

leads to further degradation of the annual budget, without achieving better performances than SRef at the monthly scale. For both inversions, this translates into a slightly larger total posterior error (2.15 and 2.20 PgC/year, respectively, compared to 2.03 in SRef). The doubling of the prior uncertainty in SE.x2 allows it to depart more from the prior and to derive better domain-scale flux estimates, both monthly and annually, but it also leads to an increase in the "added error" (lower panel of Figure 5).

### 4.2.2 Sensitivity to the error covariance structure

Inversion SC.100 and SC.500 use prior error covariance matrices constructed using respectively shorter (100 km) and longer (500 km) horizontal correlation lengths ($L_H$) than SRef. The longer covariance length in SC.500 forces the inversion to favour large-scale, low amplitude flux corrections over localized strong adjustments. Since the prior error follows a relatively homogeneous pattern, SC.500 effectively produces a better estimation of the NEE, especially in Eastern Europe where the network is sparse (Figure SI4b). The opposite happens with SC.100, which tends to concentrate the flux adjustments in the vicinity of the observation sites.

At the domain scale, the annual budgets are nearly identical in SC.100, SC.500 and SRef. However the total error reduction is lower in SC.100 and higher in SC.500, compared to SRef (respectively 0.78, 1.28 and 1.02 PgC/year), but the "added error" is larger in SC.500 (0.23 PgC/year) and lower in SC.100 (0.10 PgC/year): this confirms the hypothesis that these are aggregation errors, that can be reduced by increasing the number of degrees of freedom in the inversion (for instance by reducing the covariance constraints).

### 4.2.3 Sensitivity to the observation network density

Compared to SRef, SO.A uses only high-altitude observations (plus LMP and TTA as these were the only sites available in their region) and SO.P uses only low altitude sites. In terms of annual budget, SO.P outperforms most of the other inversions, but as for SE.3Hcst, this results from poorer flux corrections in summer rather than from a better overall reduction of the uncertainties. On the contrary SO.A leads to results very comparable to SRef at the domain scale, with a nearly identical seasonal cycle and net annual flux. The net error reduction remains however slightly better in SRef (see also Figure SI1 for the seasonal cycles of SO.A and SO.P).

### 4.3 Evolution of the fit to the observations

The comparison of the prior and posterior model fit to observations is a classical diagnostics of atmospheric inversions (Michalak et al., 2017). The inversion is expected to improve the overall fit to the observation ensemble, and a lack of statistical improvement would generally be a sign of a malfunctioning inversion algorithm. At a finer scale, analysis of when and where the representation of the observations is most improved (or degraded), can provide useful insights on the performance of the inversion (adequacy of the definition of uncertainties) and on those of the underlying transport model.

In the right panel of Figure 7, we compare the statistical distribution of prior and posterior observation fit residuals for inversion Sref. The plot confirms that the inversion leads to an overall improvement of the representation of observations, albeit modest (prior bias (model-obs): 0.2 ppm; posterior bias: 0.05 ppm; prior RMSE: 4.9 ppm; posterior RMSE: 3.75 ppm). The left panel shows the RMSE reduction at each observation site (the size of the dots is proportional to the number of assimilated observations at each site, and the color shows the net RMSE reduction). At all sites the inversion leads to improvements in the fit, but those are generally much more modest in Western Europe, which can be explained by the (coincidental) good performance of the prior in that region (see Figure 6), but also by the strong sensitivity of these sites to background concentrations. Sites in the UK and, in particular, Ireland sample very little continental air, which leaves little margin for the inversion to improve the representation of their observations.

The center panel of Figure 7 compares the RMSE reduction of inversion SRef to that of the other OSSEs. The best performances are logically achieved by SE.x2, which can depart much more from its prior than the other inversions. On the other hand, SC.100 systematically underperforms the ensemble, which is consistent with its poorer flux error reduction. In general however, the reduction of misfits are very similar and are not good indicators for the quality of the optimized fluxes.

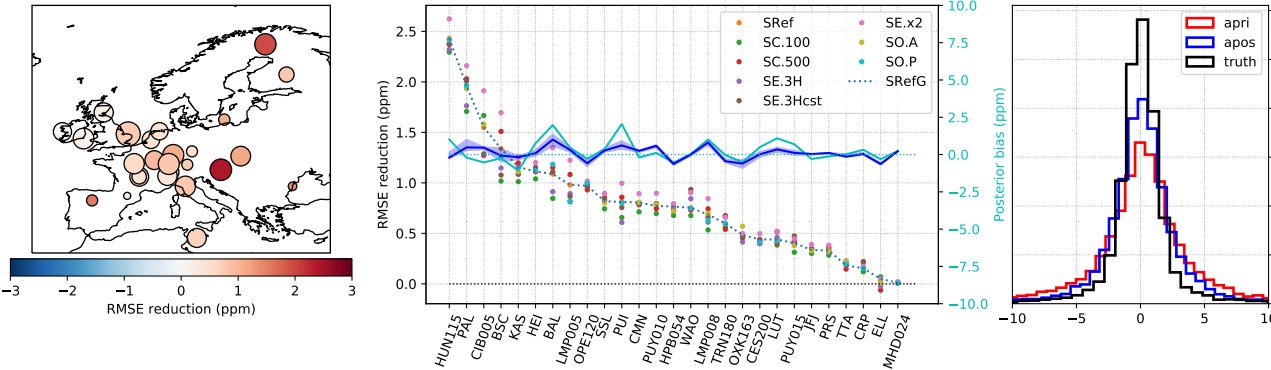

**Figure 7.** Left: Map of the observation sites in Sref, with the area of the dots proportional to the number of assimilated observation at each site, and the color proportional to the RMSE reduction (prior RMSE minus posterior RMSE). Center: RMSE reduction at each site, for the five sensitivity OSSEs. Right: distribution of observation residuals with the prior, posterior (SRef) and truth fluxes. The blues lines in the center plot show the prior (dashed line) and posterior (solid line) mean biases at each site (axis on the right)

.

## 5 Inversions with real observations

The OSSEs presented above neglect several complications of real inversions, in particular transport model errors (the obser-
535 vations were generated using the same transport model as the one used in the inversions). While it is not within the scope of this paper to quantify precisely these errors, we nonetheless performed a series of inversions constrained by real observations, to assess to which extent the characteristics of the inversion results identified with the OSSEs remain under a more realistic situation.

The set of inversions used here is identical to the set OSSEs, except that real observations are used and that the LPJ-GUESS
flux is used as a prior (instead of ORCHIDEE in the OSSEs). The inversion settings are reported in Table 3.

### 5.1 Posterior fluxes

The monthly and annual prior and posterior NEE are shown in Figure 8, for the reference RRef inversion and for the sensitivity tests. The inversion leads to a slight increase in the seasonal cycle amplitude, with a peak summer uptake increased by 24% in May (-0.36 PgC/month, instead of -0.24 PgC/month in the prior) and a nearly doubling of the $CO_2$ emissions in winter (+0.12
PgC/month instead of +0.07 PgC/month in the prior in November). It also leads to a delayed date for the change of sign of the net flux, both in the spring and in the autumn (the net prior flux becomes negative in March in the prior, and positive again in August, while it only becomes negative in April, and positive in October in RRef).

These monthly flux adjustments do not result in a change in the net annual flux (-0.33 PgC/year, both in the prior and in the RRef posterior). As seen when analysing the OSSEs results, the net annual budget is not well constrained by the inversions and
550 the absence of change is here purely coincidental.

In contrast to the OSSEs, the transport model error is not zero, which may explain the slightly higher sensitivity of the results to the extent of the observation network: RO.P and RO.A differ by, on average, 0.02 PgC/month, double of the average difference between SO.P and SO.A. However the overall spread of results in that second set of inversions is on the same order of magnitude to that obtained with the OSSEs, with a monthly spread ranging from 0.02 PgC/month (January and September) to 0.07 PgC/month (March and August). This indicates that the conclusions of the OSSEs regarding the robustness of the results can be generalized to these inversions with real data.

Maps of the prior and posterior fluxes, as well as the flux adjustments obtained with RRef are shown in Figure 9, for three 4-months periods. The January to April and September to December periods correspond approximately to the time of the year when a positive NEE correction is obtained by the inversion, while May to August is the period when the inversion finds increased uptake compared to the prior. While at large scales, the inversion preserves well the spatial distribution of NEE, the flux adjustment is not as homogeneous as what was obtained with the OSSEs (see also monthly flux adjustments maps in Figure SI5b).

The ensemble variability (lower row of Figure 9) is much higher than in the OSSEs in North-Western Europe (Northern France, Ireland and the UK), and in Hungary, around the Hegyhatsal observation site (see also Figure 6). In the latter case, this is mainly due to the inclusion or not of this site in the inversions (i.e. RO.A/RO.P inversions). The discrepancies in North-West Europe were already present in the OSSEs, but here with real observations the inversions additionally have to compensate for the inaccuracy of the transport model. In particular, errors in the prescribed background concentrations will have a stronger impact on the optimized fluxes in the vicinity of sites that sample predominantly background concentrations, such as the sites in Ireland and the UK. But also, observation sites downwind of large urban areas are more susceptible to be impacted by errors in the prescribed fossil fuel emissions, either because the emission scenario itself is incorrect, or the transport model resolution is too coarse to correctly represent the impact of these emissions at the observation site.

## 5.2   Reduction of the observation misfits

Figure 10 provides an overview of the model-data mismatches for RRef and, at the site level, for the sensitivity experiments. As expected, the inversion leads to a reduction in the RMSE, from 5.8 ppm in the prior to 4.8 ppm in the posterior, and to a slight reduction of the mean bias (from -0.2 ppm to -0.1 ppm). These values are slightly larger than the ones obtained in the OSSEs, which is consistent with the presence of a non-perfect transport model and boundary conditions.

At the site level, the prior biases are more variable than in the OSSEs, from -9.1 ppm at Baltic Sea (BSC) to +2ppm at Ochsenkopf (OXK). The bias corrections remain very modest at most sites (the bias even slightly increase at a few sites). The large (7.5 ppm) bias at BSC (Black Sea) is computed from a very small number of observations (17 in total, with observational errors up to 8 ppm), which have therefore very little weight in the inversion. The RMSE is generally reduced, except at ELL (Estany LLong, Spain) and OXK (Ochsenkopf, Germany), where the fit to the observations is slightly degraded. Both sites are located in relative proximity to other observation sites, with which their footprints overlap: the degradation of the RMSE results from contradictory constraints provided to the inversions by these different sites. The inversion does not have sufficient degrees of freedom to improve simultaneously the fit at all sites, and therefore degrades the fit to the OXK and ELL observations, which

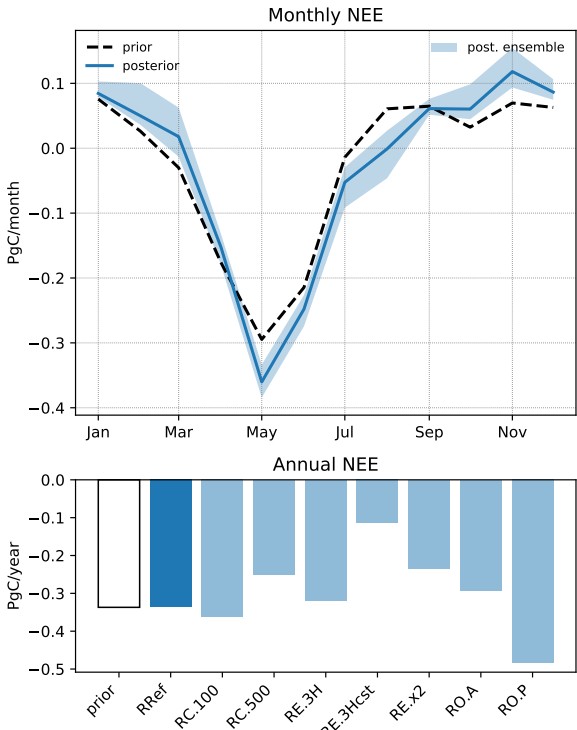

**Figure 8.** Top: Monthly prior (dashed black line) and posterior NEE (solid line: RRef; shaded blue area: ensemble spread); Bottom: Annual NEE for the prior LPJ-GUESS and the seven inversion posteriors.

have only few observations (48 and 8, respectively). The problem is common to all the sensitivity runs, and the mean posterior biases are also very similar across the inversions.

As seen with the OSSEs, a better performance in the fit to observations is not necessarily an indication of a more accurate optimized solution. The site-by-site analysis of the misfits might point to limitations of the transport operator, but a more in-depth analysis would be required, which is out of the scope of this paper.

## 6 Discussion and conclusions

We have setup an atmospheric inversion system based on an implementation of the variational inversion approach (Section 3.1) with a transport model based on an offline coupling between FLEXPART (high-resolution regional transport) and TM5 (coarse-resolution transport of the background fluxes and historical atmospheric $CO_2$ burden). The inversion setup was then tested through a series of synthetic experiments and realistic inversions, designed to verify that the inversions work as expected and to test the sensitivity of the results to typically user-defined settings. In this section we discuss separately three aspects of the paper. First the inversion results themselves, then the TM5-FLEXPART coupling and finally the LUMIA system.

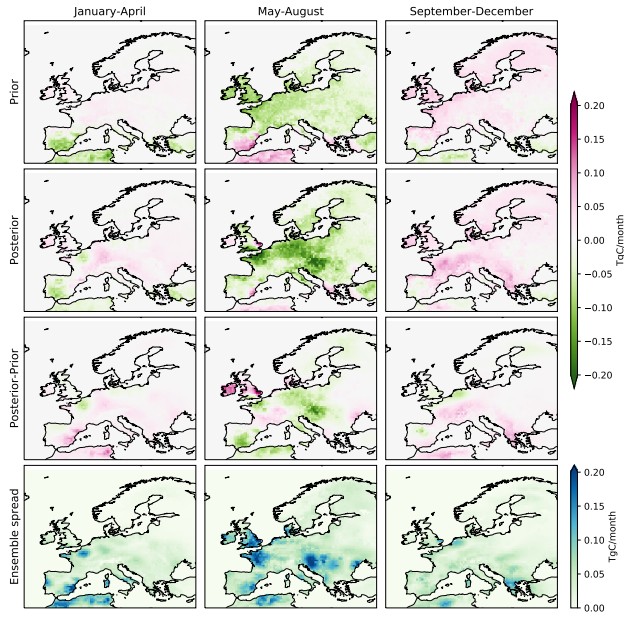

**Figure 9.** Total prior NEE (top), posterior NEE (second row) and NEE adjustment (third row) for the RRef inversion, and for three 4-months period (left to right); Lowe row: Posterior ensemble spread

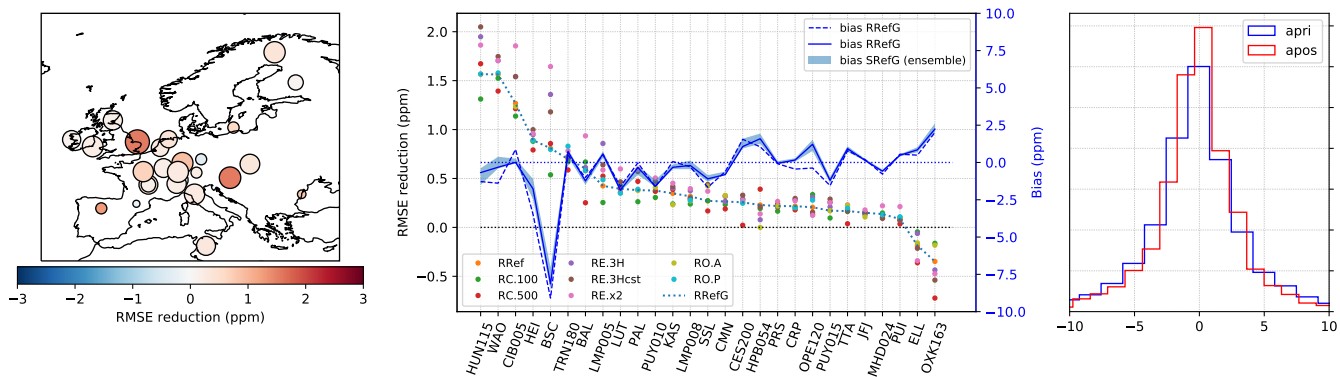

**Figure 10.** Left: Map of the observation sites in Rref, with the area of the dots proportional to the number of assimilated observation at each site, and the color proportional to the RMSE reduction (prior RMSE minus posterior RMSE). Center: RMSE reduction at each site, for the five sensitivity OSSEs, and prior (dashed blue line) and posterior (solid blue line) mean biases at each site (axis on the right). Right: Prior and posterior distribution of the observation mismatches in Rref (irrespective of the site)

.

## 6.1 Inversion approach and results

The inversion setup was designed to optimize European NEE at a monthly, 0.5° scale, based on in-situ observations from a tall tower network. The setup is intentionally simple as the focus at this stage was on developing a robust technical base and have a reference setup for future projects. The transport model is a transposition to TM5 and FLEXPART of the off-line coupling

approach developed for TM3 and STILT by Rödenbeck et al. (2009), and the optimization itself shares many similarities with existing variational inversion systems, e.g. TM5-4DVAR (Basu et al., 2013), TM3-STILT (Kountouris et al., 2018) or even PyVAR-CHIMERE (Broquet et al., 2011), which should facilitate the comparison of results with these systems. We discuss our results under the perspective of determining how well our inversion system is capable of estimating some characteristics of the $CO_2$ fluxes, and to identify directions for further improvements.

The first inversion results suggest that the inversion system is working as expected. In the OSSEs, the inversions enable on average a 40% reduction of the monthly flux error at the grid-cell scale, and the differences between the optimized fluxes obtained from different sensitivity runs are in line with what could be expected from the different settings used. However, these local error reduction can be of opposite sign, and do not always add up to a net error reduction at larger scales. In particular, while the NEE estimate is generally always improved at the monthly scale, the positive corrections in summer are much stronger than the negative corrections in winter, which results in an overall degradation of the annual NEE. Using an even month-to-month distribution of the uncertainties (SE.3Hcst inversion) leads to a more realistic annual estimate, but also to a higher occurrence of local degradations of the solution, which further complicates the interpretation of the results.

This high sensitivity of the annual NEE to the different choices of prior uncertainty show that this specific metric is not well constrained in our inversions. With further tuning, it might be possible to find a formulation of the prior uncertainties that allows OSSEs to converge towards the true annual NEE (see e.g. Kountouris et al. (2018)), but there is no guarantee that applying the same methodology in real inversions would enable to reliably determine the true NEE, since the differences between the "truth" and prior in our OSSEs don't necessarily approximate well the real error of the bottom up estimates. It may in fact remain very hard to estimate more robustly the annual NEE without introducing further constraints in the system. One straightforward way to do so would be to increase the density of observations: indeed, we see that the results gain in consistency (i.e. become less sensitive to sensitivity experiments) where the observation network is dense, which is encouraging since the number of observation sites in Europe has significantly increased compared to the data selection used in this paper.

A complementary approach could be to make a better use constraints from observations outside the domain: by definition, they cannot be accounted for directly by the regional inversion, but they were used to constrain the global inversion from which the background concentrations were extracted. By construction, the flux estimate in that global inversion is consistent with observations downwind of Europe, which isn't necessarily the case in our regional inversions (there is no constraint on the $CO_2$ concentration of the outgoing air, and therefore on the net regional flux). One may then argue that, if the global inversion provides a more reliable $CO_2$ flux estimate at the continental scale, then this large scale flux estimate could be used as additional constraint in the regional inversion. This would clarify the respective role of the regional and global inversions: the global one would estimate the fluxes at the scale of the continent and the regional one would redistribute them at finer scale. This however raises the question of the confidence level that can be given to the global inversion. Especially, in the Eastern part of the domain, there are not many observation sites and a lot of vegetation-atmosphere $CO_2$ exchanges, which the global inversion could easily misplace. This is one of the reasons why other systems haven't implemented such an approach, but we nonetheless believe that it should be explored further, in a dedicated study.

There is an ongoing debate on the net European $CO_2$ budget (e.g. Scholze et al. (2019)). While it is not the aim of this study to provide such an estimate, our sensitivity tests show why such a metric may be hard to agree on between different inversion systems.

The annual NEE is an important metric as it summarises the net impact of an ecosystem on the carbon cycle, but there are other aspects of the solution that the inversions solve for more robustly, and which are potentially equally relevant to focus on. For instance, in the OSSEs, regardless of the specific inversion setup, the posterior provides a much more realistic depiction of the seasonal cycle of NEE, and of its spatial variability. The corrections to the seasonal cycle phasing and amplitude are also very consistent across the set of inversions using real observations. This type of information is potentially very relevant when assessing the validity of flux estimates from vegetation models, and can help pinpointing specific shortcomings in these models. For instance, the consistent correction of winter NEE towards more positive values could hint to an underestimation of winter respiration in LPJ-GUESS. Note though that such a statement should be supported by a form of independent validation (such as comparisons with independent observations and with results from previous studies), which we haven't provided since the focus of the study is the validation of the inversion approach and not the $CO_2$ fluxes themselves.

Another important aspect is the distribution of fluxes at finer spatial scales. We see that the OSSEs systematically lead to some degradation of the solution in the parts of the domain that are very densely covered by the observation network, which is counter-intuitive. It may be partly because the prior was already very close to the truth in this part of the domain, which makes it difficult for the inversion to further optimize the solution, but a complementary explanation is that the system may not have sufficient degrees of freedom to adjust the fluxes to simultaneously improve the fit at all observation sites. In particular, the optimization of monthly fluxes is very restrictive. The implementation of an optimization at a higher temporal resolution will therefore be an important next step. In addition, varying the resolution of the optimization according to the density of the observation network may also help (either by varying the resolution of the optimized fluxes, or by varying the covariance lengths in the prior error-covariance matrix).

Finally, the application of the same inversion approach to real observations leads to overall smaller flux adjustments than in the OSSEs. This could be a sign that the difference between the LPJ-GUESS prior (used in this second set of inversions) and the true fluxes is smaller than that between the prior and synthetic truth in the OSSEs, but the analysis of the observation misfits reduction also point to potential site-dependent transport model errors. One of the next steps towards improving our inversions will therefore have to be a thorough assessment of the transport model biases. In that sense, the flexibility of LUMIA with regards to the transport model is particularly adapted.

## 6.2 TM5-FLEXPART coupling

The inversions rely on an offline coupling between the FLEXPART Lagrangian transport model (for regional, high resolution transport) and TM5-4DVAR for providing background concentrations. The setup replicates the 2-step scheme of Rödenbeck et al. (2009) but with different models.

A succint comparison between this "TM5-FLEXPART" transport model and TM5 itself was performed and is used as a proxy for the transport model error. It doesn't show any global bias between the two models, but a possible seasonal offset

towards the month of November (Section 3.4.1). The prescribed observation uncertainties are scaled up to account for this possible larger model error, so the impact on inversions should be limited. Nonetheless, that possible seasonal bias would need to be investigated and accounted for before deriving scientific conclusions from inversions against real observations.

The choice of the models and of that specific coupling was driven in part by the perspective of exchanges with other groups using similar setups. In the current stage, replacing the FLEXPART footprints with footprints from another similar Lagrangian transport model (e.g. STILT(Lin et al., 2003), NAME (Jones et al., 2007), etc.) or the TM5 background time series by data generated with a different model (using either the same or a different technique to estimate background concentrations at the observation sites) is straightforward and will facilitate a better evaluation of the model performance.

The Rödenbeck et al. (2009) approach means that there is no 'hard' coupling between the two models meaning that there is no risk of having to use an older version of one model because of the lack of implementation of the coupling in newer code. This, of course, also facilitates the exchange of transport model, as mentioned above.

From a practical and technical point of view, the setup presents the advantage of speed and scalability: the application of the transport operator is done independently for each observation and therefore can be distributed on as many CPUs as available. It consists, for each observation, in a very simple sequence of operations, both in the forward (Equation 9) and adjoint steps. This enables inversions to be performed in very limited (user) time (5-8 hours wall time per inversion on 24 CPUs for the inversions in this paper). On the other hand, the Lagrangian footprints and the background concentration time series need to be pre-computed, which adds to the overall cost of the inversions. The computation of footprints for 10,000 observations (approximately the number used in this study) represents 30 CPU hours (in our experience, the performance of FLEXPART depends a lot on the specific configuration and on the technical specifications of the computer performing the calculations, so this number needs to be taken with caution). The computation of the background concentrations represented 150 CPU hours (less than 24 hours of clock time, with the model parallelized on 8 CPUs). The cost of computing background concentrations could be further reduced by using "standard" optimized flux products, such as CAMS or CarbonTracker (replacing step 1 of the background concentrations computation, in Section 3.3.2).

These steps are significant overheads, but they need to be computed only once, regardless how many inversions are performed with these footprints and background concentration time series and they also contribute to the modularity of the setup (it would be easy to replace FLEXPART footprints by footprints computed with a different Lagrangian transport model, e.g. STILT or NAME). Our setup is therefore particularly adapted for conducting large set of inversions such as presented in Sections 4 and 5, which provide a better (though still incomplete) representation of the uncertainties. For comparison, in our experience, a single global TM5-4DVAR inversion with a 1° zoom over Europe (the highest resolution permitted by that model) takes on the order of 1,000 CPU hours. The computational requirements of Eulerian models such as TM5 increase rapidly with the resolution (explaining why our "background" TM5 inversion remains relatively fast to compute), which also limits our capacity to use them in high-resolution inversions (progress is being made with their parallelization, but Lagrangian models are by nature more efficient at high resolutions). One limitation of our setup, however, may be with using very large datasets of observations (such as satellite retrievals): since the computational time scales linearly with the number of observations, further developments

to limit the overall cost of such inversions would be beneficial (aggregation of observations and footprints; reduction of the number of grid points where possible; etc.).

## 6.3 The LUMIA framework: conclusions and future perspectives

We have developed the LUMIA inversion framework, and performed a first set of inversions with it. The framework is initially designed for the purpose of performing regional $CO_2$ inversions in Europe, however it is designed and developed as a flexible and adaptable inversion system, which enables the easy exchange of major components of the system, such as the transport model or the minimization algorithm, to isolate and study their impact on the inversion results. LUMIA is designed to be transport-model agnostic, i.e. it is not constructed on top of an existing model and it calls the transport model via a well-defined interface.

Technically, the inversion framework presented in this paper includes three major components: the lumia python library, which contains most of the actual inversion code in the form of independent modules; a transport component, which relies on pre-computed observation footprints and background concentrations; the inversion scripts themselves, which use the lumia library and the transport model to implement the inversion experiment.

The lumia python library defines a set of classes corresponding to basic elements of the inversion setup, e.g. control vector, gradient descent algorithm, transport model interface, observations database, etc. The library is distributed on a git server and is installable via the standard pip tool, which means it can be installed in one single command on a new computer. Although the library is developed and designed for the purpose of inversions, we have made particular effort to ensure the modularity of the code: the different modules can be imported independently and used to construct new experiments. On the short-term, this design facilitates the re-use of the code in pre-/post-processing steps of the inversions, as well as during the analysis of inversion results. On the longer term, the intention is to avoid that our initial design choices restrict the implementation of future experiments.

For this initial paper, we have performed regional $CO_2$ inversions, intentionally using a rather 'classical' inversion design to ensure comparability with other similar setups and to have a reference of comparison for future inversions, but also because it enabled us to focus on the technical robustness of the code. The transport is performed by a script which relies on pre-computed FLEXPART observation footprints, and on background concentrations pre-computed with the global coarse resolution TM5-4DVAR inverse model (although technically, nothing limits the use of alternative models to compute these footprints and background concentrations).

Although the inversion setup lacks the maturity of established systems, it offers promising computational performance and the results suggest interesting scientific questions regarding the capacity of regional inversion systems to constrain the annual budget of $CO_2$, and point to specific improvements of the inversion approach, which will be implemented in the near future, e.g. the optimization of fluxes at a higher temporal resolution. The system has also recently been used in a regional inversion intercomparison exercise centered on Europe (Monteil et al., 2020) and it has shown performances comparable to those obtained with other systems, which reinforces our confidence in the validity of the approach.

On the longer term, the aim is to use LUMIA as a platform for testing innovative inversion approaches (multiple transport models, use of satellite data, multi-tracer inversions, optimisation of vegetation model parameters (CCDAS), etc.). The code corresponding to the inversions in this manuscript is provided for the research community on https://lumia.nateko.lu.se (last visited: April 2021), and the access to the git server can be granted on demand.

*Code availability.* The LUMIA source code used in this paper as well as updates can be downloaded from the lumia website: https://lumia. nateko.lu.se (last visited: April 2021).

*Author contributions.* G.M. and M.S designed the experiments and G.M. developed the code and performed the simulations. G.M. prepared the manuscript and M.S. provided corrections and suggestions for improvements.

*Acknowledgements.* G.M. has been funded by the Swedish Research Council project 'Development of regional ecosystem-atmosphere models assimilating the ICOS data for a European-scale intercomparison of net CO2 fluxes - Eurocom' (DNR 349-2014-6576). The research is part of three Swedish strategic research areas: ModElling the Regional and Global Earth system (MERGE), the e-science collaboration (eSSENCE), and Biodiversity and Ecosystems in a Changing Climate (BECC). The computations were enabled by resources provided by the Swedish National Infrastructure for Computing (SNIC) at NSC partially funded by the Swedish Research Council through grant agreement no. 2018-05973.

We thank Michael Mischurow for providing the LPJ-GUESS net ecosystem exchange data, Philippe Peylin for providing the ORCHIDEE NEE fluxes, and Greet Janssens-Maenhout for providing the fossil fuel product. We thank the FLEXPART and TM5 developers for providing the transport models source codes. Finally, we thank all the observation data providers cited in Table 1 for providing the observations used in this paper, and we thank NOAA/GMD for collecting and distributing the data in obspack format.

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
