# Peer review of "Regional CO2 inversions with LUMIA, the Lund University Modular Inversion Algorithm, v1.0"

_Geoscientific Model Development, 2019_

## Author Comment (AC1) · 4 Oct 2019

There is an error with Figure 9 of the manuscript (it shouldn't be identical as Figure 6!). Please find below the correct figure.
* * *
[Figure]

**Fig. 1.** nnual (bar plots) and monthly (line plots) prior and posterior fluxes aggregated on the entire domain (upper row) or in smallerregions.

---

## Referee Comment (RC1) · Anonymous Referee #1 · 11 Nov 2019

This paper describes a regional flux inversion framework that is designed to have modular functionality. The performance of the system was demonstrated with a series of Observing System simulation experiments and real data experiments. They showed that the flux inversion has improved monthly mean fluxes and fitting to the observations irrespective of experimental setup in the OSSEs, but the fitting to the observations get worse at some sites with real observations. In spite of the improved monthly mean fluxes, the annual total fluxes get worse with almost all experiments. The paper did not explore ways to improve annual total flux estimate. Though this is a modeling development study, I would recommend more discussions about how to improve the system and the advantage of the regional flux inversion compared to existing flux inversion

system. Here are my detailed comments:

1. It is necessary to demonstrate that the regional flux inversion system developed here outperforms coarse global flux inversion. In this paper, TM5-4DVar is used as boundary conditions for regional flux inversions. I would recommend including discussions on the comparison between LUMIA and the TM5-4Dvar in both OSSE and real observation experiments.

2. The authors attributes the poor annual flux estimates to the larger adjustment to summer fluxes due to larger prior uncertainty. Since improving annual flux estimates is one of the major goals of regional flux inversions, I would recommend authors exploring ways to improve annual flux estimates, especially with OSSEs. In OSSEs, both true and prior fluxes are known, so specification of prior flux uncertainty can be based on the true prior flux errors. The percentage prior flux errors could be much larger during winter than during summer.

3. Validation of flux estimates from top-down flux inversion is a necessary step to assess the quality of the system. The framework described in this paper is lacking the flux validation component. A common method is to compare the posterior concentrations against independent CO2 concentrations. With high resolution regional fluxes, is it possible to use other independent observations?

4. Please add computational cost of each component of the inversion system

5. Page 24, line 516, replace "than" with "as"

6. Page 26, Line 531, replace "im" with "in".

———————————————————

---

## Referee Comment (RC2) · Anonymous Referee #2 · 25 Feb 2020

It is a good effort by the authors but I had a hard time understanding whether it is an algorithm/software oriented paper or paper completely devoted to scientific results. The authors themselves say quote:

"The inversion technique used in this study is by design not innovative (the definition of the control vector, the specification of the uncertainties, etc. replicate what has been done in previous studies (e.g. Kountouris et al. (2018)), as the aim is to have a reference setup. The scientific results are therefore at this stage limited (as it also wasn't the aim of the paper), but the analysis of the OSSEs results show that the inversions are working as expected" [So nothing new here].

If it is not the aim of the paper then why devote half the manuscript to it !!! With respect to code the authors say quote: "

"The LUMIA code is not meant to be a "key in hand" system, it target users having or willing to acquire robust understanding of inverse modelling (it is perfectly usable as a toy model for learning). We therefore do not publish the code in a public repository, but we are very open to collaborations and distribute the code on-demand. An archive of the code in it's current shape is nonetheless included as SI of this document"

So the authors do not think that their results are innovative or new, and they are not willing to publish the full code so I do not understand what is the strength of the paper? Furthermore, it looks like there code is just for their own research. Many groups have similar code and in some cases better documented fully functional codes are also publicly available. For e.g. see https://www.geosci-model-dev-discuss.net/gmd-2019-185/ and the code is also publicly available: https://github.com/greenhousegaslab/geostatistical_inverse_modeling Another one: https://www.esrl.noaa.gov/gmd/ccgg/carbontracker-lagrange/doc/index.html

Please help me understand what is new in the manuscript!!

---

## Author Comment (AC2) · 6 Apr 2020

We would like to thank the reviewers for their feedback. Based on their comments we propose to submit a revised version of the manuscript. Please also note that the paper was in review for six months, during that time there have been lots of developments on our side, and therefore we also have our own critical opinion on some aspects of the paper, which we think justifies a revision and of course would like to include in a revised version.

Below are our detailed answers to the reviewer's comments

[Figure]

**1 Reviewer 1**

We would like to thank the reviewer for their suggestions. The reviewer has made four main recommendations:

*The reviewer asks us to to demonstrate that the regional inversiouns outperform global flux inversions*

There is no simple answer to that question:

- At the continental scale, the global inversions might in fact perform better than the regional inversions. This is because there is no constraint on the CO2 concentration of the outgoing air mass in regional inversions (and therefore on the net regional flux), so flux adjustments by the regional inversions don't have to be consistent with CO2 observations outside Europe, in contrast to what happens in global inversions where there is by definition a constraint by the global atmospheric growth rate.

- The main benefit of regional inversions over global ones is the capacity to correctly assimilate observations from dense networks and/or more complex sites: the resolution of the global TM5 simulations used for producing our background concentrations is $6° \times 4°$ ( 450 km at $45°$N). This is absolutely not adapted to the density of the ICOS network in Western Europe, and it is also not adapted for assimilating data at sites that are nearby strong CO2 point sources such as urban centers. We expect the regional inversions to be more performant at such smaller sub-continental scales, but global inversions should remain more relevant at large scales.

- It would theoretically be possible to increase the resolution of the global model, but we would then encounter a performance limitation: a one-year LUMIA inversion with ICOS data typically takes 3 to 6 hours. A one-year TM5 inversion with

a 1° zoom over Europe takes up to 6 days.

We agree that the comparison between the performance of regional and global inversions is a very relevant and important topic to study, but it is complex topic and should be treated thoroughly in a dedicated study. Such a study would include a comparison between LUMIA and TM5 (but with TM5 ran at a 1° resolution over Europe because of the above mentioned limitation of a coarse spatial resolution, and initialized with the same prior as LUMIA). It should also include alternative formulations of the regional inverse problem, such as imposing a constraint on the net European C flux from the global carbon budget. Finally, it would require consequent amount of validation data.

We however agree with the reviewer that this performance comparison is an important topic and hence should be discussed more explicitly in the paper. We will reinforce that aspect in the discussion section of the revised manuscript.

*The reviewer also suggested that we test alternative parameterizations of the prior uncertainty, to attempt fitting better the "true" net annual flux (when known).*

In short: we agree with this suggestion, and we propose to include this in the revised paper. We can do it by scaling the uncertainty to the respiration and not to the NEE, which leads to less seasonal variations of the uncertainty. However, we must stress out that even with a more adequate representation of the uncertainties, there is no way to guarantee that this issue would not happen in an inversion using real observations (as it is impossible to guarantee that the uncertainty matrix is adequate). This issue is probably common to most inversions, which is why we think it is relevant to highlight it.

*The reviewer would like to see a validation of the optimized fluxes and suggests doing it via a comparison against independent CO2 concentrations.*

It is technically very feasible (and easy) if using only point observations (I.e. in-situ or flasks) for the validation, but we fear that this comparison would be mostly informative of

the local performance of the inversion and therefore would be very difficult to interpret. The possible exception would be airborne or satellite data, but this would also open many questions on the best way to represent these data in a coupled transport model. Another justification for not carrying out such a validation is that, the OSSEs provide a form of validation. At least they demonstrate that the inversion is functioning as expected, the cases where it does not lead to an improvement are explainable by an inadequate setup, and not by a malfunction of the inversion algorithm. The inversions using real observations are mainly here to give an idea of how the model behaves in a less ideal case (I.e. with transport model errors and with a (probably) more complex pattern of prior error) but we do draw conclusions on the optimized fluxes themselves.

*The reviewer asks for computational cost of the components of the inversion.*

The reviewer is entirely right, and we will add this to the revised version.

In summary, we propose for the revised version to improve the definition of the prior uncertainties in order to improve the estimation of the annual net budget in the OSSEs (and hopefully also in the inversions with real data). We also agree to extend the discussions of the two other points raised by the reviewer in the revised manuscript: 1) validation of the fluxes and 2) comparison with global inversions.

**2   Reviewer 2**

The reviewer strongly questions the interest of the paper, criticizing in particular the fact that we "are not willing to publish the full code", and that the results are not innovative or new. He/she wonders whether the focus if the paper is on the software or on the scientific results.

- *on the publication of the code*: It is wrong to say that we are not willing to publish it: the code was provided, in its entirety, as SI of the paper. However, we can

understand that the reviewer would prefer to see the code in a public repository. One reason for not doing so at the initial submission was that we were in transition from python2 to python3, and therefore the specific branch of the code that was used to produce in this manuscript was not going to be maintained. We have now completed this transition and are willing to publish the code on an online git repository, as part of the revision.

- *On the general rationale of the paper*: LUMIA was developed as a system that would allow us to perform regional CO2 inversions, and to adjust independently various aspects of the inversion system (the transport model, the optimization algorithm, the formulation of the control vector and of its uncertainties, etc.). We did not have an in-house transport model and there were, at the time when we started developing LUMIA, and to our knowledge, no generic enough inversion tools publicly available to do this (the systems pointed at by the reviewer didn't exist or were only in early stage of their development). We therefore built from scratch a new inversion system, based on a new offline coupling between TM5 and FLEXPART, and a completely new python library handling the actual inversion. The main aim of the paper is therefore to describe and publish that inversion library, including the TM5-FLEXPART coupling, so that a) we can refer to it in future studies based on this system, and b) other people can use it also.

- We chose to first focus on developing a robust technical basis (computational efficiency, "cleanliness" and modularity of the code, portability, etc.) instead of trying to directly be innovative with the inversion technique. We therefore first implemented a rather simple and classical inversion approach. This facilitated the testing and the comparison with other similar inversion systems. Therefore don't think that the lack of scientific innovation of the inversion approach is a weakness at this stage.

- The aim of the sensitivity tests is to verify that the system behaves as expected,

and also to detect weaknesses of the current approach to help us identify the components where further developments are most needed. Furthermore, our inversion setup shares many similarities with several other regional inversion systems, the limitations that we identified are likely to apply to these systems as well. We therefore think that these sensitivity test results are relevant not just for us, but more general for the inversion community as a whole and therefore we think it is relevant to include them in the paper.

Besides these clarifications, we agree that several aspects can, and should be improved. As mentioned above, we can now publish the code on a public repository. Since the submission of the manuscript the code has been migrated to python3, is now better documented, and should be easier to setup and understand for new users. Some of the text of the paper is misleading/imprecise and the clarifications written above need to be integrated as part of the revision (mostly by improving the text of introduction and discussion).

---

## Author Response (AR1)

We thank both reviewers for their comments. Because of the time between the submission and the reception of the reviews, the LUMIA code had evolved significantly. In particular, the code used in the initial submission was written for python2.7, and was therefore not meant to evolve further (which in part justified not publishing the code on a public repository). We have now fully transitioned to python3, and have setup a website and git repository for the code (with a more complete documentation). This has also lead to some changes in the structure of the code, which are reflected in the paper. In response to the reviewer's comments, we have also modified some of the sensitivity tests (see list of changes below), and have removed some aspects of the results analysis.

Below are the reviewer's feedback, with our replies in red. Note that there are some differences from the initial Author comment that we posted after receiving the reviews (due to the time taken for the revision).

Further down in this document, we provide a summary of the changes, and finally a marked-up version of the manuscript

**Reviewer 1**

This paper describes a regional flux inversion framework that is designed to have modular functionality. The performance of the system was demonstrated with a series of Observing System simulation experiments and real data experiments. They showed that the flux inversion has improved monthly mean fluxes and fitting to the observations irrespective of experimental setup in the OSSEs, but the fitting to the observations get worse at some sites with real observations. In spite of the improved monthly mean fluxes, the annual total fluxes get worse with almost all experiments. The paper did not explore ways to improve annual total flux estimate. Though this is a modeling development study, I would recommend more discussions about how to improve the system and the advantage of the regional flux inversion compared to existing flux inversion system.

Here are my detailed comments.

1. It is necessary to demonstrate that the regional flux inversion system developed here outperforms coarse global flux inversion. In this paper, TM5-4DVar is used as boundary conditions for regional flux inversions. I would recommend including discussions on the comparison between LUMIA and the TM5-4Dvar in both OSSE and real observation experiments.

We agree that the comparison between global and regional model is an interesting research question, but we feel that it would be difficult for us to do this in an appropriate way in this paper:

- At the continental scale, global inversions might in fact perform better than regional inversions, because there is no constraint on the CO2 concentration of the outgoing air mass in regional inversions (and therefore on the net regional flux), so flux adjustments by the regional inversions do not have to be consistent with CO2 observations outside Europe, in contrast this is implicitly taken care of in global inversions where there is by definition a constraint by the global atmospheric growth rate.

- The main benefit of regional inversions over global ones is the capacity to correctly assimilate observations from dense networks and/or more complex sites: the resolution of the global TM5 simulations used for producing our background concentrations is 6°×4° (450 km at 45°N). This is absolutely not adapted to the density of the ICOS network in Western Europe, and it is also not adapted for assimilating data at sites that are located close to strong CO2 point sources such as urban centers. We expect the regional inversions to be more performant at such smaller sub-continental scales, but global inversions should remain more relevant at large scales.

- It would theoretically be possible to increase the resolution of the global model, but we would then encounter a performance limitation: a one-year LUMIA inversion with ICOS data typically takes 5 to 8 hours. A one-year TM5 inversion with a 1° zoom over Europe takes up to 6 days.

The comparison between global and regional inversions is therefore a rather complex topic, and we think that it should be done in a dedicated study. In particular, the TM5 inversion should use a higher resolution over Europe (it can run at 1° resolution), and the LUMIA inversion would benefit from additional developments, with in particular a higher temporal resolution of the optimization and more fine-tuning in the data selection.

2. The authors attributes the poor annual flux estimates to the larger adjustment to summer fluxes due to larger prior uncertainty. Since improving annual flux estimates is one of the major goals of regional flux inversions, I would recommend authors exploring ways to improve annual flux estimates, especially with OSSEs. In OSSEs, both true and prior fluxes are known, so specification of prior flux uncertainty can be based on the true prior flux errors. The percentage prior flux errors could be much larger during winter than during summer.

We have changed some of the sensitivity experiments to go in that direction. In particular, the new experiments SE.3H, SE.3Hcst and SE.x2 explore different distributions of the prior uncertainty throughout the year, which, unfortunately, do not really solve the problem. The fact that this metric is so sensitive to the inversion setup suggests that the surface observations used in our inversions do not appear to provide sufficient constraints to resolve the annual budget.

We discuss this aspect further in Section 6.1: one approach would be to add further constraints on the inversion (e.g. by adding more observations into the system, by adding constraints on the CO2 concentration of the air exiting the regional domain, or by adding the annual budget as a control vector variable). We feel however that this question should be addressed via a specific, dedicated study (it is not specific to our model, but is rather a general issue with regional inversion systems).

3. Validation of flux estimates from top-down flux inversion is a necessary step to assess the quality of the system. The framework described in this paper is lacking the flux validation component. A common method is to compare the posterior concentrations against independent CO2 concentrations. With high resolution regional fluxes, is it possible to use other independent observations?

It can technically be done, the same way than with a global inversion, but there are limitations that make it impractical: To our knowledge, there is no comprehensive observation dataset that could be used for validation purpose (for example, we are not aware of a European equivalent of the HIPPO campaigns that were made in the US). Also the interpretation of such comparison could be risky: the observation network is not homogeneous, and the performance of the inversion can vary widely from one region to the other, which would make a comparison with independent data difficult to interpret. A potentially more promising approach would be the use of satellite data, but this would also open many other questions, regarding how to best model them in a coupled system.

Instead, the validation of the results relies on three aspects: Firstly, the OSSEs provide a first demonstration that the inversion is functioning as expected. Secondly, performing an (ideally large) ensemble of sensitivity runs (including the use of alternative transport model, which hasn't been shown here) helps assessing the robustness of the results (and the limitations of the inversion system). Finally, the participation of inter-comparison exercises is a key aspect of the evaluation of the results.

4. Please add computational cost of each component of the inversion system

The computational cost of the LUMIA inversions is briefly mentioned in Section 6.2. That cost (5-8 hours on 24 CPUs) is dependent on the number of observations, and does not include the cost of computing the footprints and background concentrations:

- the cost of computing the footprints depends on the number of observations (it takes ~1 CPU hour to compute 50 footprints with FLEXPART, for our regional domain), however, each footprint needs to be computed only once.

- likewise, the TM5-4DVAR inversion from which the backgrounds are extracted takes approximately 24 hours to compute, for one year, but the background extraction itself (one TM5 forward run) takes only a few minutes (regardless the domain size and the number of observation sites). The inversion needs to be computed only once, and even that could be skipped in the future, by using the fluxes from a "routine" TM5 inversion, such as CarbonTracker (for CO2) or CAMS (for CH4).

The 5-8 hours therefore correspond only to the marginal cost of each additional inversion performed. By comparison, a one year TM5 inversion with a 1x1 zoom over Europe takes 5 to 7 days to complete. Our background TM5 inversion is not a good base for comparison, as it is really optimized for speed, at the detriment

of accuracy of the posterior fluxes (it needs to fit well the observed spatial gradients of CO2, but the accuracy of the optimized fluxes is secondary as those are not kept, therefore we can reduce the number of iterations).

5. Page 24, line 516, replace "than" with "as"

The whole sentence has been modified

6. Page 26, Line 531, replace "im" with "in".

The whole sentence has been modified

**Reviewer 2**

It is a good effort by the authors but I had a hard time understanding whether it is an algorithm/software oriented paper or paper completely devoted to scientific results.

As explained in the paper, the inversion algorithm is not new, neither are the transport models used (TM5 and FLEXPART), however this particular combination has never been implemented before, and the software used to implement that coupling (LUMIA) is new. The experiments presented are designed to test the performances and limitations of the system, rather than to derive actual scientific conclusions on the European NEE (we do not discuss those). We think that it is a necessary stage before the use of the system for production of scientific results.

The authors themselves say quote:

"The inversion technique used in this study is by design not innovative (the definition of the control vector, the specification of the uncertainties, etc. replicate what has been done in previous studies (e.g. Kountouris et al. (2018)), as the aim is to have a reference setup. The scientific results are therefore at this stage limited (as it also wasn't the aim of the paper), but the analysis of the OSSEs results show that the inversions are working as expected" [So nothing new here].

If it is not the aim of the paper then why devote half the manuscript to it !!!

We agree with the reviewer that some parts of the results section were leaning too much towards scientific results, which was not in the scope of this paper. We have therefore largely revised the results sections. We have slightly reduced the number of figures, and have removed the analysis by regions, which makes Section 5 shorter (Inversions with real observations). However, Section 4 has increased a bit in length, in response to reviewer's 1 comments, so the overall length of the results analysis did not change.

With respect to code the authors say quote: "

"The LUMIA code is not meant to be a "key in hand" system, it target users having or willing to acquire robust understanding of inverse modelling (it is perfectly usable as a toy model for learning). We therefore do not publish the code in a public repository, but we are very open to collaborations and distribute the code on-demand. An archive of the code in it's current shape is nonetheless included as SI of this document"

Maybe there was a confusion here: the code was not made available on a public (git/svn/ftp) repository, but the (entire) code was published as SI of the manuscript, with a documentation. We have however improved on that aspect, and we now have a proper website (https://lumia.nateko.lu.se) and git repository, from which the code can be downloaded. The repository also contains several pages of documentation/tutorial, in the form of jupyter notebooks.

So the authors do not think that their results are innovative or new, and they are not willing to publish the full code so I do not understand what is the strength of the paper?

The aim of the paper is to publish and document the new LUMIA inversion system, also in order to provide a reference for future research building on it.

Furthermore, it looks like there code is just for their own research.

We indeed developed the code primarily for our own research (as do most research groups?), but we do not think that this would be a sufficient reason not to document it and test it correctly.

Many groups have similar code and in some cases better documented fully functional codes are also publicly available. For e.g. see https://www.geosci-model-dev-discuss.net/gmd-2019-185/ and the code is also publicly available: https://github.com/greenhousegaslab/geostatistical_inverse_modeling Another one:https://www.esrl.noaa.gov/gmd/ccgg/carbontracker-lagrange/doc/index.html

These systems did not exist, or were only in early stage of their development when we started working on LUMIA (and they are not directly comparable to it either). And again, the fact that other systems with a similar purpose exist should not be a sufficient justification for disqualifying our work: replication of results is an important part of research.

Writing and keeping up to date an extensive documentation such as the one of CT-Lagrange is a time investment that is only justified if the user-base is large enough. There are indeed many other research groups developing internally similar types of codes, but not making them publicly available. This is not a good practice, but discouraging authors to publish on the basis that the documentation is insufficient leads to this situation.

The compromise that we had found was to distribute the code along with the manuscript, and to encourage interested users to contact us (which is always a good practice when using the research code of someone else). For this revised version, we have however made the code and documentation available online.

Please help me understand what is new in the manuscript!!

**Summary of the changes**

**Major revisions**

- Sections 2.2 and 2.3 have been merged and completely re-written, to reflect the changes in the code. Figure 1 (the flow diagram of the inversion) has also been edited to reflect these changes.

- Section 3.4.1 has been simplified (only the "gaussian" approach to construct the horizontal covariance length is now described, the "exponential" approach has been left out, as it did not add value to the paper). On the other hand, the section now lists two other approaches used to construct the prior uncertainty vector.

- Section 4 has been almost entirely re-written, and the selection of inversions that we show has been modified:

  - Inversions SB and SG have been removed; All the inversions now use covariance matrix constructed from Gaussian covariance decay functions (so SRef is now identical to the previous SG). The inversion SE.3H, SE.3Hcst and SE.x2 are new, and test different construction of the uncertainties (in response to reviewer's 1 comment). All the inversion names have been modified (to reflect the grouping of sensitivity tests: inversions SE.3H, SE.3Hcst and SE.x2 test the impact of the prior uncertainties; SC.100 and SC.500 test the impact of changes in the prior error covariances; SO.A and SO.P test changes in the observation network.

  - The sub-structure of the section has been modified as well: we now first analyse the reference inversion, in terms of monthly, annual and pixel-scale fluxes and errors, and then look at the impact of the sensitivity runs. The analysis by regions has been removed.

  - Figure 5 has been modified, to show the sensitivity runs as an ensemble, for clarity and conciseness. The individual results can still be found in SI.

  - Figure 6 has been removed.

  - Figure 7 (now 6) has been highly simplified. The information from the previous plots can still be found in SI Figures, but is not discussed in the paper.

- Section 5 has been largely simplified, and the inversion selection has been modified as well:

  - The inversions RE.3H, RE.3Hcst and RE.x2 have been added (the selection of inversions is now similar in Section 4 and 5). The other inversions have been renamed following the naming convention used for Section 4.

  - The regional analysis has been removed, and as a consequence Figure 9 (now 8) has been simplified. The entire section has been made shorter and the text is now focusing only on the differences between OSSEs and real simulations. The sensitivity tests are not discussed individually, but only as an ensemble.

- The discussion Section 6.1 has been completely rewritten, in part to account for some of the comments of reviewer 2, and in part to reflect the changes in the results section.

- The content of the conclusion has now been moved to a new discussion section 6.3 (The LUMIA framework: conclusions and future perspectives), and enriched, in part also to account for the comments of reviewer 2.

**Minor revisions**

- Minor edits in the abstract to better reflect the aims of the paper

- Section 2.1 renamed to "theoretical background", and small textual changes.

- A paragraph has been added at the end of Section 3.2, to clarify the difference between observation operator and transport model.

- Figure 11 (now 10) has been replotted, since an error was found on the original figure (the bias instead of the RMSE reduction was shown in the left panel). The right panels of Figures 11 (now 10) and 8 (now 7) have also been edited, for clarity.

[revised manuscript text omitted]

---

## Referee Report (RR1)

**Review of "Regional CO2 inversions with LUMIA, the Lund University Modular Inversion Algorithm, v1.0" by Monteil and Scholze**

This manuscript (MS) describes a regional flux inversion framework that was designed to have modular functionality. The performance of the system was demonstrated with a series of Observing System simulation experiments and real data experiments. It is a model description MS. I reviewed this manuscript by following the GMD model description papers guideline, <a href="https://www.geoscientific-model-development.net/about/manuscript\_types.html">https://www.geoscientific-model-development.net/about/manuscript\_types.html</a>. This MS has relatively deep level of details about the LUMI framework. The authors made the code and documentation available online so the community can easily use it for their work. I will skip those requirements this MS meets and only list those ones it does not meet. The unmet requirements are listed below, and I would not recommend this MS for further publication if they are not carefully addressed.

Technical details are not completely clear to me. In section 3.2.1, the authors mentioned that FLEXPART was driven with ERA-I. In section 3.2.2, they used TM5-4DVar posterior CO2 mole fraction as boundary condition. So, the transport models used in the domain of interest and outside were different. Have the authors studied the conservation of mass in the boundaries? The mismatch can cause the biases in the flux estimation. Additionally, in the later text, it said that FLEXPART was driven with the TM5 transport. Please clarify and correct it throughout the text.

Back to section 3.2.1, the authors only ran FLEXPART even days backward. It seems too short to me, but I am not positive. This will be determined by the weather system in the regions. How did the authors come up this number? Do the users have the flexibility to change this number for their work?

In section 3.2.2, the description of TM5-4DVAR is misleading. It was only used to provide the 3D CO2 mole fraction fields. The description of the posterior fluxes is unnecessary. Additionally, the description of CarbonTracker CT2016 is not correct. CT2016 used CASA-GFED for prior instead of SibCASA. Please check other components and correct it if anything is inaccurate. https://www.esrl.noaa.gov/gmd/ccgg/carbontracker/CT2016/CT2016\_doc.php

Lack of the evaluation against standard benchmarks or observations. The MS provided OSSE and real data experiment. The results are hard to justify because no reference is used. There are quite some inversion results publicly available, such as CarbonTracker posterior fluxes at global scale, CarbonTracker-Lagrange system at regional scale, etc. The comparison against those state-of-art system are necessary to justify the suitability of LUMIA.

Another related point, the description of the difference/improvement of LUMIA compared to those inversion systems would be helpful and appreciated for users to choose a framework that is more appropriate for their research. The authors could also compare their results with the flux measurements if the observations are available, such as AmeriFlux, Fluxnet, etc.

Last but not least, the authors should use "ensemble" more carefully. Technically, the experiments designed only leads this work to a sensitivity study. The transport error is a big deal in inversion. The authors didn't even offer an experiment that uses different transport models in this study. Thus, the experiments designed are far from ensemble, and the RMSD over these experiments are far from model uncertainty.

Some other special concerns below.

**Abstract:**

I would not include any citation in the abstract. The authors should articulate the problem clearly instead of simply giving citations in the abstract.

The authors used the first two paragraphs for the background information in abstract following by introducing LUMIA starting with "some of these new topics". What exactly are the topics the authors are referring to? Please clarify them.

L30: LUMIA is an inversion framework. It has nothing to do with "predict the evolution"

L39: I would not use "robust estimates". Reducing the uncertainty is the nature of the Bayesian inversion, but it doesn't mean the final estimation is robust.

Figure 2: please use height above ground level instead of altitude. It would be more informative by using height so that readers (like me) would have sense in the mixing depth, etc.

---

## Author Response (AR2)

We thank both reviewers for their comments on the paper, which have helped us improving the paper compared on several aspects.The main changes are the following:

- The abstract has been revised, in response to comments from reviewer 4 and from the editor

- Section 3.2 had clearly caused confusions for both reviewers, especially reviewer 4. We have split it in two and revised large most of the text to improve its readability and provide more complete details.

- We have expanded the discussion, following comments from reviewer 3 mainly

- We have tried to improve the quality of the English writing,

We provide below a point by point answer to the reviewers (our comments are in green).

**Reply to reviewer #3**

We thank the reviewer for his/her very constructive comments, which have helped to improve the paper. We agree with almost all the suggestions, and have been to integrate most of them. Our point by point reply is given below, with our replies in green.

This paper describes LUMIA, a new modular regional inversion framework to estimate CO2 fluxes. The framework is demonstrated with pseudo and real CO2 observations used to estimate net ecosystem exchange over Europe, and the sensitivity of the results to various inversion settings. I believe that the paper is a worthwhile contribution. I think the balance between description of the framework, presentation of results and sensitivity studies is about right to highlight some important aspects of the framework. Some of the responses to Reviewer 1 would be worth including in the paper but as far as I can see are not currently there. Some of the inversion terminology and concepts should be explained more clearly for readers that are not already very familiar with them. Also, the paper needs some editing to improve the English (I have suggested some corrections below). I recommend publications after attention to the following comments:

Response to Reviewer 1: Some of the discussion in the response to Reviewer 1 doesn't come through in the revised paper and would be valuable to add. For example, Comment 1 - add the paragraph that mentions the outgoing air mass

We had very briefly mentioned this in section 6.1 (l564-549), but we agree that it could be interesting to develop that comment further, therefore we have expanded that discussion (still in Section 6.1).

, and the paragraph on the performance limitation.

We have expanded the discussion around performance in Section 6.2.

I think it is worth suggesting ways to improve the annual estimate (as already mentioned in the response to Reviewer 1, comment 2, on air exiting the domain, and putting a constraint on the annual budget).

See our first reply above. We added some of the elements of our reply to reviewer 1 directly in the paper, but we definitely cannot close the discussion on this in this paper: there are many possible implementations of such an additional constraint and there are actually also arguments against doing it. We have tried to briefly synthesize them but the topic would deserve a dedicated study itself.

The authors say the poor annual flux estimates are a general issue with regional inversions, has the problem been discussed or solved elsewhere that could be referred to?

We actually participated (and co-lead) a model intercomparison exercise on this exact topic (EUROCOM), now published in ACP (Monteil et al., 2020), and this was one of the main outcomes. But the EUROCOM study was submitted after this GMD manuscript and cites the original GMDD paper, therefore we are not sure if we can cite it.

Other than that, there are many inversion studies for North America, but there the problem is simpler since it is easier to have a reliable estimate of the $CO_2$ concentration of both the air entering and the air leaving the domain (delineated by oceans on both sides). In Europe, there is an ongoing debate on the carbon balance of the continent, largely traceable to uncertainties in the Eastern part of the domain. This is summarized in Scholze et al., 2019, however this manuscript doesn't specifically focus on regional inversions so we don't think it's a good reference for that statement.

Mention the cost of computing the footprints in the paper.

We now provide a more complete overview of the computational costs in Section 6.2.

line 32: "For future climate simulations, the only available options is through "direct" (bottom-up) modelling ..." - this may be confusing to readers. Only option for what? The whole sentence could be improved to be more informative.

We have edited and enriched the sentence, we hope it is clearer now.

line 38 - 'inverse approaches provide robust estimates of total fluxes at large scales ...' - perhaps temper this with 'can provide'. The Gurney paper is a global study, this should be pointed this out. The example in this study shows that total fluxes are not always robustly estimated on large scales by inversions. 'On the contrary' is out of place here.

We agree with the reviewer that the phrasing wasn't ideal and we have revised the sentence (along with the paragraph above).

It was, however, not completely incorrect: bottom-up models really don't provide much constraints at all on the net annual C flux. There is a very large diversity of ecosystem models, and these models are constantly evolving, so it is difficult to make too general statements, but if anything, these models tend to be  explicitly calibrated to result in  a reasonable global C flux estimate.

Note also that the Gurney paper is quite old: it serves well our purpose here but lots of progress has been made with atmospheric inversions since its publication, both in terms of development of the observation networks and in terms of accuracy of the transport models.

line 57-58 - why are there a lack of options for cross-validation? Perhaps also briefly explain what cross-validation is.

We were referring here to the possibility to compare with results from different studies, which is more difficult with regional inversions since the study domains are less likely to overlap. We realized that it was neither a very strong nor important argument, so we have simply removed it.

line 72-73 - explain foreground and background concentration

This is explained in further details in Section 3.3, and it would not be easy to explain it concisely in the introduction. Instead we have replaced "foreground" by "regional" and "background" by "global".

I think it would help readers relatively new to the area if a couple of sentences were added to explain how footprints come from and relate to the transport model, that we can take H(x) as equiv. to Hx, and how inversions can use either the footprints or the full transport model and adjoint.There are elements of this information scattered around, but something fairly early on to draw it all together, maybe around line 87, I believe would help.At line 146, H(x) equiv to Hx comes after both H(x) and Hx have been used separately without being clear how they relate, so this information should come earlier than line 146.

We have added a sentence clarifying the equivalence between $H(\mathbf{x})$ and $\mathbf{Hx}$ in Section 2.1.

As part of wider revisions of Section 3.3 (mostly to address reviewer 4 comments), we also have added a sentence in Section 3.3 to explain why footprints can be used instead of a full transport model.

line 89 - 'In the simplest cases, the system can be solved for x analytically, but most often inversions use the Bayesian inference approach' - I believe that a simple case be solved analytically and also use the Bayesian approach? The sentence makes it sound like it can't.

Indeed, this was misleading. We have replaced "analytically" by "directly".

line 96 - 'of each departure from each prior control variable xib and from each observation yj' - the each ... each ... each doesn't sound good. Can the sentence be improved?

We have fixed this.

line 129: The scripts folder is not mentioned above with the other folders, lumia and transport. Are there any other folders? Can all folders be mentioned together?

In fact "scripts" here should just be "GMDD". We have fixed that mistake.

Fig 1 - the text is very small in the printed copy. Flux files, and Flux pre-processor - are these just for the prior fluxes? If so, could you add 'Prior'?

We realize that the figure was too small, and have now made it larger.

"Fluxes" is not equivalent to "prior" (the fluxes are, in our case, in 3 categories (biosphere, ocean, fossil), and at a 3-hourly temporal resolution, while the prior only contains the monthly biosphere flux). Depending on what is implemented in the "Model/Optimized Interface" box, the prior could also contain non-flux variables (for instance, some inverse modellers optimize the boundary condition).

line 145-149 - number these steps, to match step numbers at line 162

We have replaced the bullet points by numbers.

Fig 2 - could add latitude and longitude numbers around the grid

We have added the lat/lon axis to the figure, and also slightly modified it to make it more readable.

line 162 - add lower case omega after 'vector' to emphasize what quantity is referred to here.

Done.

> line 179 - 'the inversion adjusts an offset to the prior, high temporal resolution fluxes' - I would put that sentence before equation 6, as I think it is easier to understand in words first, then eqn 6.

We have revised the whole section, which should hopefully be much clearer now.

line 193 - sometimes the word 'footprints' is used, other times 'response functions'. It is good to use both terms when the concept is first introduced, but then stick to one of the other.

The reviewer is right, we have removed the references to "response function", especially since the meaning of the two terms can be a bit ambiguous in literature.

line 212 - number the steps

Done.

line 211 - 'inversion-derived CO2 fluxes' - could add the word global again here to emphasize that they are from a global inversion

We have removed that sentence, since it wasn't very clear and is explained better and more detailed just below.

line 222 - is subtraction of foreground concentrations from total ones the same as running the model with only fluxes outside the regional domain?

No, it's not the same. In reality, CO2 fluxes within the regional domain affect the background (the air that leaves the domain eventually travels around the world, and re-enters at some point). If we just set the fluxes to zero within the regional domain, we would lose that impact (if, as our simulations show, Europe is a net carbon sink, then nullifying the fluxes in Europe would over time lead to an excess CO2 in the background concentrations).

One simple thought exercise is the following:

- Make the hypothesis of zero fluxes and concentrations everywhere, except for a single emission pulse at some place within the regional domain, at the beginning of the simulation. Eventually the CO2 emitted in that pulse will leave the domain, travel around the world and re-enter the domain, as background concentrations.

- If, in order to calculate the background, we just nullify the fluxes within the domain, then the concentrations will just remain zero forever. On the contrary, subtracting foreground concentrations doesn't affect in any way the concentration of the air entering the domain (since the transport of that CO2 is computed by the "full" TM5 simulation).

We acknowledge that the approach is not very intuitive at first. As part of revisions of Section 3.2 (now 3.3), we have clarified the distinction between foreground and background, and the respective roles of the two transport models. We have also added two sentences in Section 3.3.2 to justify the interest of this approach. We stress out the need to refer to the Rödenbeck et al., 2009 paper for further information (as it is dedicated to that topic).

line 231 - 'accounted for' - need to be more specific about how they are accounted for. Same at line 322.

We have replaced the first occurrence by "transported". We have further clarified in Section 3.2 and 3.3 (both previously 3.2) that the transport model "transports" fluxes in several flux categories, therefore the second occurrence should be clear enough.

line 232 - what is 'It'? The global inversion?

line 236 - this refers to the global inversion, right? Add 'global' after 'The'?

We have made the sentence more explicit, to address these two remarks.

line 246 - 'for each observation site' - in the regional domain?

Yes, we have made this explicit.

line 282 - 'very contained' - is there a better word to use here?

We have replaced "very contained" by "very small"

Table 1 - Could add to the caption that set 'A' is mainly high altitude sites, and set 'P' includes only low altitude sites.

Indeed, this was missing, we have now added it.

line 365 - I don't understand 'absolute next monthly flux'

The word "next" shouldn't have been here and we removed it.

Table 3 is a good description of the cases, I would refer to it a bit earlier in the text, perhaps before the tests are listed (e.g. 'In the sensitivity tests (Table 3), we vary ...').

We have followed the reviewer's suggestion.

Table 3 caption refers to the letters R and B that are no longer relevant.

Indeed, we have removed this sentence.

line 391 - 'In total, the absolute prior error slightly exceeds 3 PgC...' what quantity is this referring to, and what are the units?

line 394 - units PgC/month?

The unit was really meant to be PgC (it is the error accumulated over the entire simulation). For clarity, and since the simulations are one year long, we replaced the unit by PgC/year.

line 396 - 'this results in a strong degradation of the annual estimate' - presumably this happens because the prior was already a good estimate of the annual NEE. If the prior had been worse at the annual scale this may not have happened. Similarly at line 551.

Yes, but it goes against the commonly held assumption/intuition that a reduction of the errors would necessarily lead to an improvement of the flux estimates at large scales (here, temporal).

We have added a sentence around line (previously) 396 to stress this.

line 397 - begin a new paragraph to move from discussing temporal to spatial.

Done.

Fig 5 caption - Mention at the start that this is for the OSSE. Be clear that the 'ensemble' is the set of sensitivity tests (also in Fig 6 and section 4.2 and elsewhere).

We have edited the caption to follow the reviewer's requests.

Section 4.2 repeats earlier text.

Indeed, some information was given twice, in sections 3.5 and 4.2. We have removed it from section 4.2.

Fig 7 - blue and cyan are very similar in the middle plot. Is 'posterior bias' the right y-axis label for the right plot, why not 'distribution of residuals'? Making the blue line in the right plot a lighter blue would make it easier to distinguish from the black line.

"Posterior bias" is in fact the label of the right axis of the middle plot. We have now made it more obvious by coloring that axis in the same color, and increasing the distance between the right and center plots. We have improved the figure. We have also changed the colors in the right plot for better readability.

line 579 - 'model calibration effort' - is that calibration of the transport model or terrestrial model?

Here we were referring to the calibration of the transport model. But the phrasing wasn't very good. We have replaced the sentence by "a thorough assessment of the transport model biases" (the aim is not really to "calibrate" the models, but to correctly detect and account for systematic model errors).

An OSSE could be done with one atmospheric transport model used to calculate the pseudo observations and a different one in the inversion.

Yes, this would be very relevant, however, as mentioned in further details in our reply to reviewer 4, this falls outside the scope of this paper.

line 586 - 'possible seasonal offset towards the month of October' - be more specific.

We have moved the reference to Section 3.3.1 just after that sentence, since it is explained in that section. Again, we didn't emphasise too much the transport model validation as it was not the aim of the paper (and more data would be needed to do it), so it is hard to be much more specific here.

Have other regional European inversions also had trouble constraining the annual budget of CO2?

There is an ongoing debate on the net European CO2 budget, summarized in our recent EUROCOM paper (Monteil et al., 2020). We have added a reference to that debate in Section 6.1.

Check formatting in some refs - Peters et al, NOAA 2019, van der Laan.

The NOAA reference was indeed incorrect, it pointed to the CH4 version of GLOBALVIEW. We have corrected some other formatting references.

line 767 - update Rayner et al (2018) reference to 2019, ACP

Done.

Minor corrections/English: [...]

We thank the reviewer for pointing these out, we have corrected them (when they were still relevant).

**Reply to reviewer #4**

We thank the reviewer for his/her comments and we have tried to account for them in our revisions as much as we could. However some of the critique is difficult to take into account, as it is clear that the reviewer misunderstood some aspects of our work.

In particular, the reviewer didn't understand the coupling strategy between our regional and global transport models, questions it and calls it "misleading", based on incorrect assumptions. This misunderstanding is of course partly our responsibility: we should have described our work better, and we have revised some of the methodology sections to avoid such misunderstandings from future readers. However, the core of that critique is on aspects of the transport models coupling that simply replicate another study (Rodenbeck2009, which we cite and refer to multiple times): we provide sufficient information to allow others to replicate our work, and we have now added some sentences explaining why we chose that specific coupling strategy, but we don't think that we should discuss in details its merits or limitations, as an entire publication is already dedicated to it.

The second major comment ("lack of evaluation against standard benchmarks or observations") is difficult to take into account: the reviewer criticizes the "lack of comparison with standard benchmarks or observations". However, such standard benchmarks simply don't exist (the reviewer seems to consider that CarbonTracker can be used as a benchmark, which we strongly disagree, for reasons developed in the detailed reply below). We do provide a relatively extensive set of inversions (OSSES + inversions with real data), which prove beyond reasonable doubt that the system works as intended. The performance comparison against other systems is an interesting point, but we think it is unreasonable to ask us to address it in a model description paper.

This manuscript (MS) describes a regional flux inversion framework that was designed to have modular functionality. The performance of the system was demonstrated with a series of Observing System simulation experiments and real data experiments. It is a model description MS. I reviewed this manuscript by following the GMD model description papers guideline, https://www.geoscientific-model-development.net/about/manuscript_types.html. This MS has relatively deep level of details about the LUMI framework. The authors made the code and documentation available online so the community can easily use it for their work. I will skip those requirements this MS meets and only list those ones it does not meet. The unmet requirements are listed below, and I would not recommend this MS for further publication if they are not carefully addressed.

Technical details are not completely clear to me. In section 3.2.1, the authors mentioned that FLEXPART was driven with ERA-I. In section 3.2.2, they used TM5-4DVar posterior CO2 mole fraction as boundary condition. So, the transport models used in the domain of interest and outside were different.

No, it's not like that. TM5 does not just provide a boundary condition at the edge of the domain, it is also used to transport that boundary condition to the observation points within the domain. So in fact, both models are used within the domain: FLEXPART is used to compute the transport of "foreground" fluxes, while TM5 is used to calculate the boundary condition and to transport it to the observation points (this is one of the reasons we prefer to talk of "background" and not of "boundary condition").

The approach was already explained throughout Sections 3.2 and 3.2.2 (Equation 5 and Figure 3 in particular). However, to further clarify it, we have split that section into two, and largely

> revised it. The respective roles of the foreground and background transport models are explained in Section 3.3.

> Our setup replicates the coupling scheme of Rödenbeck et al., 2009 . We provide enough information in our manuscript to fully replicate our setup, but we refer to Rödenbeck et al., 2009 for further information, and in particular for a discussion on the details of that coupling technique.

Have the authors studied the conservation of mass in the boundaries? The mismatch can cause the biases in the flux estimation.

> A consequence of the coupling strategy is that there is no exchange of $CO_2$ mass between the two transport models, so there cannot be a bias specifically related to the issue of mass conservation.

Additionally, in the later text, it said that FLEXPART was driven with the TM5 transport. Please clarify and correct it throughout the text.

> The reviewer is wrong, we have nowhere in the manuscript stated that FLEXPART was driven by TM5 transport.

> Both models are driven by ERA-Interim data. In the case of FLEXPART, this was clearly stated in the original manuscript (Section 3.2.1, line 198 of the original manuscript). For TM5, that information had somewhat disappeared during the revision of the manuscript, we have re-added it (in Section 3.3.2).

Back to section 3.2.1, the authors only ran FLEXPART even days backward. It seems too short to me, but I am not positive. This will be determined by the weather system in the regions. How did the authors come up this number?

> The footprint value generally peaks a few hours before the observations, and then rapidly diminishes as the FLEXPART particles leave the domain or travel higher up in the atmosphere. After seven days, the value of the footprints (i.e. dy/dx, with y the concentrations and x the fluxes) is zero or near zero for the vast majority of the observations.

> That seven days cutout time is just one of many other factors that affect the transport model error, and we expect other factors, such as the driving meteorology or the definition of the surface layer in FLEXPART, to have much more importance. In this paper, it has little to no consequences for our results (in the OSSEs, by construction, there is no transport model error).

Do the users have the flexibility to change this number for their work?

> Yes. It is just one FLEXPART parameter, and we do not distribute FLEXPART along with LUMIA, so it is up to the users to configure FLEXPART (or whichever other model they use to generate their footprints). A user is not tied to use FLEXPART in LUMIA.

In section 3.2.2, the description of TM5-4DVAR is misleading. It was only used to provide the 3D $CO_2$ mole fraction fields.

> No, it wasn't, and we do not think that there was anything "misleading" in Section 3.2.2. The section describes exactly what we did:

> 1. First, we performed a global inversion, using observed data both within and outside our regional domain (see the manuscript for full details).

2. Second, we run a modified TM5 forward run, which uses the fluxes computed in the first step to compute both the "total" CO2 mole fraction time series at the location of the observation sites, and also the "foreground" CO2 mole fraction, corresponding to the fluxes within the European domain.

3. Finally, we obtain the "background" concentrations time series by deducing the "foreground" from the "total".

In summary, TM5 was first used in step 1 to conduct a global inversion, and then in step 2 to compute the "foreground" mole fraction.

We have revised Section 3.2.2 (now 3.3.2) to avoid such misunderstandings, and have strengthened the reference to the Rödenbeck et al., 2009, study, which describes this coupling in much further details.

The description of the posterior fluxes is unnecessary.

We do not describe the TM5 posterior fluxes….

Additionally, the description of CarbonTracker CT2016 is not correct. CT2016 used CASA-GFED for prior instead of SibCASA. Please check other components and correct it if anything is inaccurate. https://www.esrl.noaa.gov/gmd/ccgg/carbontracker/CT2016/CT2016_doc.php

The reviewer is right, we have corrected the reference.

Lack of the evaluation against standard benchmarks or observations. The MS provided OSSE and real data experiment. The results are hard to justify because no reference is used. There are quite some inversion results publicly available, such as CarbonTracker posterior fluxes at global scale, CarbonTracker-Lagrange system at regional scale, etc. The comparison against those state-of-art system are necessary to justify the suitability of LUMIA.

Inversion results from well established systems such as CarbonTracker cannot be used as a benchmark for evaluating other models: 1) they have their own shortcomings and can also perform poorly on some metrics, without a clear possibility to detect that poor performance (e.g. where/when there are only few observations, or few other models to compare); 2) such comparisons are not straightforward to interpret. Given the differences between the inversion setups (prior, observations, uncertainties, transport model, inversion approach, etc.), we cannot expect the results to be identical, so what type of differences would fall in- or outside the acceptable range? According to which metric(s) should the systems be compared?

On the contrary, with the OSSEs we have, by construction, a "perfect" reference for evaluating our inversion results (the known truth). Of course, it neglects complications of "real" inversions, in particular the transport model error, which is why we also provide a set of inversions using real data. This is not enough to tell how well our system performs in comparison to others (but that's not the aim of the paper), but it is enough to ensure that it works as expected, and to provide good insights on how the results should be interpreted.

Finally, a comparison of results obtained by LUMIA and other systems has actually already been done and published in a separate study (Monteil et al., 2020). To avoid cross-referencing we cannot cite it as it was submitted after the current manuscript (and cites it), but it contains some of the comparisons that the reviewer requests.

Another related point, the description of the difference/improvement of LUMIA compared to those inversion systems would be helpful and appreciated for users to choose a framework that is more appropriate for their research.

There are many other systems, with different degrees of maturity, it would be hard to make an objective comparison. Also because we are obviously much more familiar with our own system than we are with others.

Furthermore, the strengths of LUMIA lie not in its unique features (the inversions presented in this manuscript are, by choice, very classical), but in its modularity: it would, for instance, be relatively easy to develop an EnKF module and replicate the CT-Lagrange setup. If one is just interested in running CT-Lagrange-like inversions, then it would be a much better idea to choose CT-Lagrange directly, but if one wants to study the impact of using a variational system vs. an EnKF one, then LUMIA would be a good choice because of its modularity (however, this specific test is not one we plan to conduct in the near future).

The authors could also compare their results with the flux measurements if the observations are available, such as AmeriFlux, Fluxnet, etc.

Once again, as mentioned earlier and in our general reply, the aim of this paper is to describe the system, demonstrate that it works, but not to provide a performance evaluation.

More fundamentally, it is difficult to compare directly flux measurements from eddy-covariance sites with flux estimates from atmospheric inversions: the first ones are (at best) representative of a specific ecosystem, while on the contrary, atmospheric inversions (at least ours) optimize the total NEE in each model grid cell (and in fact in larger regions, since, because of the imposed flux error covariance structures, neighbouring grid cells are not optimized independently): this comparison can be done and would be interesting, but as a dedicated study.

Last but not least, the authors should use "ensemble" more carefully. Technically, the experiments designed only leads this work to a sensitivity study. The transport error is a big deal in inversion.

The word 'ensemble' is ambiguous: it can either mean an ensemble of experiments with different models or an ensemble of experiments with one model but different conditions (boundary or initial conditions, model parameter settings etc). The latter is employed in an EnKF set up similar to CT-Lagrange and which the reviewer refers to as 'sensitivity study'.

Nevertheless, we agree that the transport model error is an important aspect in inversions. That is exactly one of the motivations for designing LUMIA: having a modular inversion system where the impact of transport model errors can easily be tested by replacing the transport model. However, before we reach that step, we must start by developing the inversion system itself and this is exactly what the current manuscript is about.

The authors didn't even offer an experiment that uses different transport models in this study.

We do not offer such an experiment because it is clearly out of scope of the current manuscript as explained in the reply above.

Thus, the experiments designed are far from ensemble, and the RMSD over these experiments are far from model uncertainty.

The reviewer seems to imply that we claim having covered the full uncertainty space of the fluxes with our simulations. As mentioned above, we recognize that the word can be seen as ambiguous, and have replaced some of the occurrences of the word "ensemble" by "group" or "set", but the word "ensemble" has multiple meanings, and we don't think that the paper is ambiguous on how we constructed our "groups of inversions" and on how we interpret them. We clearly mention that these are sensitivity experiments (l565 of the original manuscript).

> The computational cost of inversions prevents from doing a formal statistical ensemble, yet, we maintain that the distribution of results from our sensitivity tests provide at least a qualitative estimate of the uncertainties (of course, limited to the uncertainty due to the parameters tested).

Some other special concerns below.

Abstract: I would not include any citation in the abstract. The authors should articulate the problem clearly instead of simply giving citationsin the abstract.

The authors used the first two paragraphs for the background information in abstract following by introducing LUMIA starting with "some of these new topics". What exactly are the topics the authors are referring to? Please clarify them.

> We have revised the abstract to address both comments above.

L30: LUMIA is an inversion framework. It has nothing to do with "predict the evolution"

> It is a context sentence, which is not directly referring to LUMIA...

L39: I would not use "robust estimates". Reducing the uncertainty is the nature of the Bayesian inversion, but it doesn't mean the final estimation is robust.

> See our reply to reviewer 3 on this.

Figure 2: please use height above ground level instead of altitude. It would be more informative by using height so that readers (like me) would have sense inthe mixing depth, etc.

> Height above ground is provided in Table 1. Altitude is much more informative since some of the sites are located on the top of a mountain (Jungfraujoch and Weybourne are both sampled at 10 m height above ground, yet they aren't exactly comparable with Jungfraujoch at an altitude of 3570m and Weybourne of 20m).

---

## Author Response (AR3)

We thank the reviewer for this second round of review. The reviewer has two minor comments, which we answer below (in red):

2nd round review on Monteil and Scholze "Regional CO2 inversions with LUMIA, the Lund University Modular Inversion Algorithm, v1.0"

The revision has improved the manuscript in terms of English and technical description. Some of my concerns on the original submission have been carefully addressed.

The other reviewer and I raised the questions related to the terminology "background" and "foreground". So, it's good to see that this confusion got clarified in the revision. In the response to us, not just once, the authors pointed out that some of the studies were not cited because they were submitted after this manuscript. I highly recommend the authors adding those citations to the final submission, which allows them to justify the application of LUMIA.

We have added a sentence in the conclusion, pointing to that paper.

When I said that "comparing other existing systems" in the last round, I meat that it would be appreciated for the authors to emphasize the limitations and strengths of the LUMIA framework compared to other existing systems so that one can select a system that fits their research goal better.

We understand the request from the reviewer, but as already explained in our previous reply, we don't think that this is something we can provide, at least not in a very meaningful way: LUMIA is intentionally not a system with a very well-defined set of features: the work presented in this paper is representative of the possibilities of the system at the time when the paper was initially submitted, but not of the current or future possibilities. The same goes to a large extent for the other systems, so a comparison, succinct or detailed, would not actually be very informative for the readers (and may even be misleading).

The main strength of LUMIA is its modularity: the code is compact, flexible and portable, which makes it easy to implement new features or to embed it in other projects. This is described at lengths in Section 2 and in the last section of the discussion/conclusions. These qualities are (probably) shared by some of the other recent inversion systems, and these systems may have qualities that LUMIA doesn't have, but we don't think that it is the right place for offering a comparison (and the other model description papers don't generally do that either).

Those points above are minor critiques. There is no need from my end to have another round review after the modification.